# Estimating return periods for extreme events in climate models through Ensemble Boosting

Luna Bloin-Wibe<sup>1,\*</sup>, Robin Noyelle<sup>1,\*</sup>, Vincent Humphrey<sup>1,2</sup>, Urs Beyerle<sup>1</sup>, Reto Knutti<sup>1</sup>, and Erich Fischer<sup>1</sup>

**Correspondence:** Luna Bloin-Wibe (luna.bloinwibe@env.ethz.ch)

## Abstract.

10

With climate change, extremes such as heatwaves, heavy precipitation events, droughts and extreme fire weather have become more frequent in different regions of the world. It is therefore crucial to further their physical understanding, but due to their rarity in both observational and climate modeling samples, this remains challenging. For numerical simulations, one way to overcome this under-sampling problem is Ensemble Boosting, which uses perturbed initial conditions of extreme events in an existing reference climate model simulation to efficiently generate physically consistent trajectories of very rare extremes in climate models. However, it has not yet been possible to estimate the return periods of these simulations, since the conditional resampling alters the probabilistic link between the boosted simulations and the underlying original climate simulation they come from.

Here, we introduce a statistical framework to estimate return periods for these simulations by using probabilities conditional on the shared antecedent conditions between the reference and perturbed simulations. We validate this framework with a simple red-noise process and find the typical time scale at which one could expect to sample stronger extremes. This is then applied to simulations of the fully-coupled climate model CESM2: first for a pre-industrial control simulation, and then in present-day conditions, where, as an example, we estimate the return period of the record-shattering 2021 Pacific Northwest heatwave to be 2500 [2000-4000] years. Our evaluation of the method shows that return periods estimated from Ensemble Boosting are consistent with those of a 4000-year control simulation, while using approximately 6 times less computational resources. We thus outline the usage of Ensemble Boosting as an efficient tool for gaining statistical information on rare extremes. This could be valuable as a complement to existing storyline approaches, but also as an additional method of estimating return periods for real-life extreme events within a climate model context.

## 20 1 Introduction

Extreme weather events, or phenomena that occur at the tails of the climatological distribution, can have devastating impacts on ecosystems, human life, settlements, and infrastructure. In recent years, climate change has caused the frequency of extremes such as heatwaves, heavy precipitation events, drought and fire weather to increase (Ranasinghe et al., 2021; Seneviratne et al.,

<sup>&</sup>lt;sup>1</sup>Institute for Atmospheric and Climate Science, ETH Zurich, Zurich, Switzerland

<sup>&</sup>lt;sup>2</sup>Federal Office of Meteorology and Climatology MeteoSwiss

<sup>\*</sup>These authors contributed equally to this work.

2021). Additionally, the non-stationary climate that arises through climate change also means that record-shattering heatwaves are on the rise, which can pose challenges for adaptation, since communities are more vulnerable to extremes they have not yet witnessed (Fischer et al., 2021).

In particular, heatwaves have become not only more common, but also more intense and long-lived compared to the pre-industrial climate, a change which is projected to continue with climate change (Meehl and Tebaldi, 2004; Rahmstorf and Coumou, 2011; Thiery et al., 2021; Seneviratne et al., 2021). Several impactful summer heatwaves have been observed in the last decades: examples include the 2003 European heatwave, claiming approximately 70,000 lives (Robine et al., 2008), the month-long 2010 Russian heatwave (Otto et al., 2012) and the unprecedented Pacific North-West (PNW) heatwave of 2021, that broke observational records by more than 5°C (Bartusek et al., 2022; Malinina and Gillett, 2024; McKinnon and Simpson, 2022; Neal et al., 2022; Overland, 2021; Philip et al., 2022; Schumacher et al., 2022; White et al., 2023). Studying such heatwaves is societally relevant, due to their significant socio-economic and ecological impacts (Gourdji et al., 2013; Dunne et al., 2013) – in particular the high mortality tolls such heatwaves can incur (Robine et al., 2008; Vicedo-Cabrera et al., 2021).

Summer heatwaves in the mid-latitudes are associated with persistent anticyclonic flow anomalies that can give rise to prolonged anomalies of temperatures (Perkins, 2015; Horton et al., 2016; Barriopedro et al., 2023). These anomalies of temperature occur through three physical mechanisms: horizontal advection of warmer air from neighboring regions, adiabatic warming by subsidence within the anticyclone, and diabatic fluxes such as increased radiation due to the clear-sky conditions and sensible heat fluxes from the ground caused by reduced soil moisture (Pfahl and Wernli, 2012; Miralles et al., 2014). The relative importance of these mechanisms can vary greatly from one region to another (Röthlisberger and Papritz, 2023).

Despite the societal relevance of – and significant research on – heatwaves and their dynamics, their quantitative study remains challenging due to limited sample sizes. Extreme events are rare by definition, and it is therefore difficult to obtain accurate climatological results concerning their dynamics. Several methods have been developed to overcome this sampling challenge. For example, climate model Large Ensembles constitute a brute-force approach to the problem of sampling extreme events, and have been used to describe heatwave dynamics quantitatively (Suarez-Gutierrez et al., 2018; Schaller et al., 2018). However, the sampling density necessary to adequately represent events such as the PNW heatwave cannot be achieved even with these computationally expensive large ensembles (Fischer et al., 2023).

Another way in which extremes can be studied is through Extreme Value Theory (Coles et al., 2001). Extreme Value Theory is based on the existence of asymptotic results of the block-maxima and peak-over-thresholds distributions of any random variable. For example, generalized extreme value (GEV) distributions are often used to extrapolate probabilistic information on block maxima from the sample at hand (Philip et al., 2020; Cooley, 2013). However, such estimations can be problematic. Using a large ensemble of a climate model, Zeder et al. (2023), for example, showed that return period estimates of temperature extremes are systematically overestimated in short records. Additionally, while process-based covariates can provide some dynamical insight into heatwave drivers, this approach remains limited compared to the information available directly from fully-coupled climate models (Zeder and Fischer, 2023).

Therefore, there has been a recent push for methods that can generate climate simulations of extremes more efficiently than by producing ever larger climate model ensembles. One such tool is the application of rare event algorithms to climate models

(Wouters and Bouchet, 2016; Plotkin et al., 2019; Webber et al., 2019; Yiou and Jézéquel, 2020; Gessner et al., 2021; Finkel and O'Gorman, 2024). In particular, Ragone et al. (2018) used the Giardina-Kurchan-Tailleur-Lecomte (GKTL) (Giardinà et al., 2006; Giardina et al., 2011) algorithm to clone simulations that perform well with respect to a defined score function to generate new, extreme, simulations. This process is repeated in intervals of a given resampling time; at each step, new simulations are given weights based on their parent score, which allows for a retracing of steps to calculate the probability of exceeding thresholds of time averaged quantities.

65

75

By design, the GKTL algorithm is well suited for extremes that persist in time, since time needs to elapse between the repeated cloning and evaluation steps. However, its use is limited when studying shorter extremes, like week-long heatwaves or daily precipitation extremes. In comparison, Ensemble Boosting, proposed by Gessner et al. (2021), is more suited for simulating short, very intense events. It consists in perturbing the initial conditions of extreme events selected from an already existing climate model simulation, thus efficiently creating alternative, potentially substantially more extreme versions of a given extreme event in a parent simulation. However, in doing so, it breaks the probabilistic link between the resulting simulations and the climate model large ensemble they come from. Finding unconditional return periods directly through the boosted simulations is thus not straightforward. Therefore, Ensemble Boosting is typically used within a storyline framework (Fischer et al., 2023; Lüthi et al., 2024), an approach that seeks to complement probabilistic confidence statements with plausible, episodic information on representative case studies (Shepherd et al., 2018).

A key difference between the GKTL algorithm and Ensemble Boosting is that in the latter case, the perturbation is performed in anticipation of the extreme: this means that there is no guarantee that the boosted simulation will be at least as extreme as the parent event, thus breaking the necessary assumption for the weighted probability calculation. To overcome this, Finkel and O'Gorman (2024) showed that a chained conditional probability calculation based on the Subset Simulation framework (Au and Beck, 2001) could be used to estimate probabilities for ahead-of-time resampled simulations. However, this calculation is based on a resampling method that is methodologically distinct to Ensemble Boosting and can be difficult to apply in typical simulation conditions of a climate model.

In this paper, we show how probabilities and return periods of extreme events found using the Ensemble Boosting resampling method can be estimated. In essence, we apply conditional probabilities — to calculate the unconditional probability of a boosted simulation, the unconditional probability of parent events selected for Ensemble Boosting is combined with the conditional probability of parents and boosted simulations given Ensemble Boosting. We additionally show, with a simple red-noise process, at which time scale Ensemble Boosting is expected to sample more extreme events.

The method is evaluated by generating extreme heatwaves for the region of the Pacific Northwest (45–52°N, 119–123°W) with the Community Earth System Model 2.1.2 (Danabasoglu et al., 2020), first with pre-industrial (PI) control simulations as a proof of concept, and then with present anthropogenic forcing conditions. The 2021 PNW heatwave is used as a case study: by boosting analogue simulations under current climatic conditions, we attempt to find the return period of this record-shattering heatwave within the model. We thus illustrate how we can both simulate the physical mechanisms and estimate the probability of very extreme, unseen events with a climate model.

The paper is organized as follows. Section 2 presents the theoretical framework of the Ensemble Boosting estimator for low probability events, showing that it is unbiased and can reduce the relative error compared to a naive estimator. In particular, we validate our approach with a simple 1D red-noise process. Section 3 presents the results for the pre-industrial control simulation with sensitivity tests, before applying the insights gained to the 2021 PNW heatwave in a present world context. In Section 4, the gain in computational resources, the choice of optimal parameter settings and the distribution of the very tail extremes are discussed. Finally, conclusions are presented in Section 5.

## 2 Methods

| Symbol                     | Formal definition                                                                                     | Description                                                                                       |
|----------------------------|-------------------------------------------------------------------------------------------------------|---------------------------------------------------------------------------------------------------|
| PNW                        |                                                                                                       | Pacific North-West, region of interest in the study                                               |
| TXx5d                      |                                                                                                       | Yearly summer max. of daily max. temperature anomalies with a 5-day running mean                  |
| Z500                       |                                                                                                       | Geopotential height at 500 hPa                                                                    |
| N                          |                                                                                                       | Number of simulation years in the reference climate model simulation                              |
| $N_{ m lead}$              |                                                                                                       | The number of lead times used to generate boosted simulations                                     |
| $N_{ m batch}$             |                                                                                                       | The number of boosted simulations per lead time                                                   |
| $N_{ m parent}$            |                                                                                                       | The total number of simulations in the Parent Ensemble                                            |
| $N_b$                      | $N_{	ext{lead}} \cdot N_{	ext{batch}} \cdot N_{	ext{parent}}$                                         | The total number of simulations in the Boosted Ensemble                                           |
| T                          | $\{T^n \mid n = 1, 2N\}$                                                                              | Set of TXx5d in the reference climate model simulation                                            |
| $T_b$                      | $\{T_b^m \mid m = 1, 2N_b\}$                                                                          | Set of TXx5d in the Boosted Ensemble                                                              |
| $T_{ m ref}$               |                                                                                                       | User-chosen TXx5d threshold. If $T^n \geq T_{\text{ref}}$ , it is selected to the Parent Ensemble |
| $T_{ m ext}$               |                                                                                                       | $TXx5d \ge T_{ref}$ from the Boosted Ensemble                                                     |
| t                          |                                                                                                       | Lead time at which Ensemble Boosting is performed                                                 |
| $AC_t^0$                   | $\{ \mathbf{X}_t \mid T_{t=0} \ge T_{\text{ref}} \}$                                                  | Set of antecedent conditions at $t$ of all members of the Parent Ensemble                         |
| $\mathrm{AC}^{\epsilon}_t$ | $\{\mathbf{Y}_t \mid \exists \mathbf{X}_t \in AC_t^0, \ \mathbf{X}_t - \mathbf{Y}_t\  \le \epsilon\}$ | Set of antecedent conditions that differ from $AC_t^0$ only by a perturbation $\epsilon$ at $t$   |
| ${\mathbb P}$              |                                                                                                       | Theoretical probability                                                                           |
| $\hat{p}$                  |                                                                                                       | Probability estimator                                                                             |
| E1                         |                                                                                                       | The most extreme parent event in the 30-member Large Ensemble                                     |
| $	ilde{E1}$                |                                                                                                       | The most extreme parent event in the 100-member Large Ensemble                                    |

**Table 1.** Definitions and descriptions of important quantities used in this study.

#### 100 2.1 Resampling low probability events

The return period of a heatwave reaching temperature T over a certain region is classically defined as the inverse of the probability  $p_T$  of exceeding the temperature T. The latter can be estimated as the empirical frequency of occurrence:

$$\hat{p}_T = \frac{1}{N} \sum_{n=1}^{N} \mathbb{1}(T^n \ge T),\tag{1}$$

where  $(T^n)_{1 \le n \le N}$  are N independent observations of the temperature T and  $\mathbbm{1}$  is the indicator function equals to 1 if  $T^n \ge T$  and 0 otherwise. Since for each observation n,  $\mathbb{E}(\mathbbm{1}(T^n \ge T)) = p_T$ ,

$$\mathbb{E}(\hat{p}_T) = p_T,\tag{2}$$

which makes this estimator unbiased. Furthermore, using the independence of each observation, its variance is  $\mathbb{V}(\hat{p}_T) = \frac{p_T(1-p_T)}{N}$ . Therefore, for  $p_T \ll 1$ , the relative error of this estimator  $\hat{p}_T$ ,  $\mathbb{RE}$ , is:

$$\mathbb{RE} := \frac{\sqrt{\mathbb{V}(\hat{p}_T)}}{\mathbb{E}(\hat{p}_T)} \simeq \frac{1}{\sqrt{p_T N}}.$$
 (3)

This shows that the relative error of the naive estimator increases as  $p_T$  decreases, i.e. when the temperature T becomes more extreme. In other words, the problem of obtaining precise climatological results for events reaching extreme values of T comes from under-sampling such events for a small number of observations N.

Ensemble Boosting addresses this problem by resampling the most extreme events of an already existing climate model simulation. In the following, we call these events the Parent Ensemble. The resampling is done by perturbing the antecedent conditions of the events inside the Parent Ensemble. Since the perturbations are of a relative amplitude of  $10^{-13}$  and are only performed once, physically consistent simulations, hereby called the Boosted Ensemble, can be generated. All members of the Boosted Ensemble are samples from a distribution biased towards the upper tail of the distribution of T (see Appendix Figure A1) and, in particular, could sample events that are more intense than any present in the Parent Ensemble.

115

125

Figure 1 illustrates the Ensemble Boosting algorithm. A detailed description of the algorithm employed is presented below 20 for temperature anomalies, but can be generalized to any quantity of interest that one wants to maximize or minimize:

- 1. A reference climate model simulation, spanning N years, is used to select the Parent Ensemble. Here, we consider the yearly summer maximum of daily maximum temperature anomalies with a running mean of 5 days (hereby denoted by TXx5d), forming a set of temperatures  $T := \{T^n \mid n = 1, 2...N\}$ , with n indicating the different simulation years. All years whose  $T^n$  exceeds a user-chosen temperature threshold  $T_{\text{ref}}$  are selected to form the Parent Ensemble. The total size of this ensemble is denoted by  $N_{\text{parent}} \leq N$ . An illustration of the reference climate model simulation can be seen in Figure 1a, and an example of an event in the Parent Ensemble can be seen in Figure 1b.
- 2. Since the objective of Ensemble Boosting is to generate a heatwave that is similar to, but substantially more intense than the parent event, each parent event in the Parent Ensemble is perturbed ahead of the parent event's peak. The number of days between the perturbation and the parent peak is called the *lead time* and denoted by t. Because of the chaotic nature

Figure 1. Illustration of the Ensemble Boosting algorithm for heatwaves. 1) The reference climate model simulation spanning the period 1801-1850. Each dot represents the yearly summer maximum of daily maximum temperature anomalies with a running mean of 5 days (TXx5d). 2) One parent event of the Parent Ensemble: the heatwave itself is represented as a function of lead time, defined here as the number of days before the parent event's heatwave peak (solid black vertical line). The y-axis is the 5-day running mean daily maximum temperature anomaly (Tx5d anomaly). 3) Boosted simulations (orange) of the parent event shown in 2), perturbed at a lead time of  $(\mathbf{c}) - 16$  days,  $(\mathbf{d}) - 12$  days and  $(\mathbf{e}) - 7$  days. The perturbation lead time is highlighted by a black dashed vertical line. At each lead time, a "batch" of 100 boosted simulations are generated.

of the model, the deviations between the boosted simulations and the parent event will initially grow exponentially with time (Lorenz, 2006). Figure 1c,d, e show boosted simulations that have been perturbed at different lead times. Here, we see how the lead time influences the temperatures in the Boosted Ensemble: simulations perturbed at -16 days in Figure 1c have diverged for too long, thus reverting back to the underlying climatology. Therefore, only a few boosted simulations reach the parent TXx5d. Simulations perturbed at -7 days, on the other hand, as seen in Figure 1e, have not diverged enough for any boosted simulation to deviate substantially from their parent. Figure 1d, at -12 days, illustrates an ideal lead time, i.e. where the spread is neither too small nor too large, and a large portion of boosted simulations exceed the parent TXx5d. We propose an estimation of this "optimal lead time" in the case of a red-noise process below. The difference between the climatological distribution and the distribution of the boosted simulations at this lead time is shown in Appendix Figure A1.

3. The perturbations are performed on the 3D specific humidity field Q at each grid cell, to obtain the perturbed specific humidity field  $\tilde{Q}^m$  for each boosted member m:

$$\tilde{Q}_{i,i,k}^{m} = Q_{i,j,k}^{m} \left( 1 + 10^{-13} R_{i,i,k}^{m} \right), \tag{4}$$

for a given boosted simulation n and a given grid cell i, j, k.  $R^m_{i,j,k}$  is a random term drawn from a uniform distribution between  $-\frac{1}{2}$  and  $\frac{1}{2}$ . Each newly generated offspring simulation is then run for 21 days, and the maximum TXx5d,  $T^m_b$ , is considered. While a systematic test has not been implemented, we do not expect the choice of variable to influence results, since the perturbation stays within numerical noise limits. Both specific humidity and temperature have been used in previous studies, with comparable results (Gessner et al. (2021), Gessner et al. (2022), Gessner et al. (2023)).

4. For each lead time and each parent, new simulations are generated in batches of size  $N_{\text{batch}}$ , corresponding to different realizations of the perturbed field  $(\tilde{Q}^m)_{1 \leq m \leq N_{batch}}$ . The total number of simulations in the Boosted Ensemble  $N_b$  is therefore:

$$N_b = N_{\text{lead}} \cdot N_{\text{batch}} \cdot N_{\text{parent}},$$
 (5)

where  $N_{\rm lead}$  is the number of lead times where perturbations are performed.

145

150

Computing the probability of exceeding an extreme temperature T naively from simulations of the Boosted Ensemble using Equation 1 leads to a probability estimation conditional on the initial conditions at lead time t of the parent events. To recover the unconditional, i.e. climatological, probability we use the framework of Subset Simulation (Au and Beck (2001)), where an iterative chain of conditional probabilities is used to estimate the probability of simulated low-probability events. Here, we illustrate how this framework can be used with a deterministic climate model, with perturbations ahead of time as in Finkel and O'Gorman (2024), and for the Ensemble Boosting setup, which generates batches of boosted simulations, but here stops after one perturbation iteration.

The probability of exceeding the threshold temperature  $T_{\rm ref}$  — used to select events for the Parent Ensemble — can be estimated using the naive estimator of the reference climate model simulation (Equation 1), since there are, by definition, events in the reference climate model simulation that exceed  $T_{\rm ref}$ . This also means that all parent events have in common that the dynamical conditions of the climate model system in the days leading up to the peak were such that the temperature  $T_{\rm ref}$  was reached.

One can thus define a set that includes the antecedent conditions at lead time t of all the parent events:

165

190

$$AC_t^0 := \{ \mathbf{X}_t \mid T_{t=0} \ge T_{\text{ref}} \}, \tag{6}$$

where  $\mathbf{X}_t$  is the state vector of the climate model at lead time t, which encompasses all the degrees of freedom of the system. Here,  $\mathbf{X}_t$  is associated with a temperature  $T_{t=0} := T(\mathbf{X}_{t=0})$  in the location of interest. The set  $\mathrm{AC}_t^0$  is imperfectly sampled in a simulation with a finite length, but we assume that the antecedent conditions of the selected parent events still represent a good enough sample. This is a key assumption and it will be further discussed in the Results and Discussion sections. The exact conditions constituting a representative sample are unknown, since we do not know the function giving the probability to reach an extreme given the current state — the so-called committor function (see Miloshevich et al., 2023). Additionally, if we did, it may not be transferrable across different regions. Instead, temperature at  $t_0 = 0$  is used as a first order approximation to determine the right antecedent conditions (at lead time  $t 

Figure 2. Schematic illustration of the antecedent condition set  $AC_t^{\epsilon}$  and the stricter  $AC_t^0 \subset AC_t^{\epsilon}$ . The dark blue line shows the delineation between  $AC_t^{\epsilon}$  and  $AC_t^0$ , i.e. the condition that at t=0,  $T=T_{\rm ref}$ . Only simulations that exceed this requirement (the Parent Ensemble) are selected from the reference climate model simulation, and are shown in blue diamonds. Other events in the reference climate model simulation are not depicted. The boosted simulations are shown as orange dots. Since the boosting perturbation is performed ahead of the parent peak, there is no a priori guarantee that at t=0,  $T_b^m \geq T_{\rm ref}$ , and the orange dots can therefore be on either side of the delineation.

$$\mathbb{P}(T \ge T_{\text{ext}}) = \mathbb{P}(T \ge T_{\text{ref}}) \frac{\mathbb{P}(T \ge T_{\text{ext}} \mid AC_t^{\epsilon})}{\mathbb{P}(T \ge T_{\text{ref}} \mid AC_t^{\epsilon})}.$$
(9)

Here,  $\mathbb{P}(T \geq T_{\mathrm{ref}})$  is the probability of reaching  $T_{\mathrm{ref}}$  in the reference climate model simulation,  $\mathbb{P}(T \geq T_{\mathrm{ref}} \mid \mathrm{AC}_t^{\epsilon})$  is the probability of reaching  $T_{\mathrm{ref}}$  in the Boosted Ensemble, and  $\mathbb{P}(T \geq T_{\mathrm{ext}} \mid \mathrm{AC}_t^{\epsilon})$  is the probability of reaching  $T_{\mathrm{ext}}$  in the Boosted Ensemble. In other words, the unconditional probability of exceeding the threshold is equal to the product of the probability of exceeding the threshold given one has entered the set, times the probability of entering the set. Note that since the condition  $T_{\mathrm{ext}} \geq T_{\mathrm{ref}}$  is necessary to derive Equation 9, we can only use it to find probabilities for boosted simulations where  $T_{\mathrm{ref}}$  is exceeded.

All terms in this equation can be estimated with the climate model simulations at hand: the probability of reaching  $T_{\text{ref}}$  can be approximated to its frequency of occurrence in the reference climate model simulation, while the probability of reaching  $T_{\text{ref}}$  and  $T_{\text{ext}}$  given  $AC_t^{\epsilon}$  can be approximated to their respective frequencies in the Boosted Ensemble. Formally this can be written as:

$$\hat{p}_{T \ge T_{\text{ext}}} := \hat{p}_{T \ge T_{\text{ref}}} \frac{\hat{p}_{T \ge T_{\text{ext}} | \text{AC}_t^{\epsilon}}}{\hat{p}_{T \ge T_{\text{ref}} | \text{AC}_t^{\epsilon}}} = \left(\frac{1}{N} \sum_{n=1}^{N} \mathbb{1}(T^n \ge T_{\text{ref}})\right) \frac{\frac{1}{N_b} \sum_{m=1}^{N_b} \mathbb{1}(T_b^m \ge T_{\text{ext}})}{\frac{1}{N_b} \sum_{m=1}^{N_b} \mathbb{1}(T_b^m \ge T_{\text{ref}})}.$$
(10)

The estimator  $\hat{p}_{T \geq T_{\text{ext}}}$  will hereby be referred to as the *boosting estimator*. As shown in Noyelle (2024) and derived in Appendix Section A, this estimator is unbiased, if the following assumptions are made:

- 1. the term  $\hat{p}_{T \geq T_{\text{ref}}}$  is independent from the ratio  $\frac{\hat{p}_{T \geq T_{\text{ext}} | AC_t^e}}{\hat{p}_{T \geq T_{\text{ref}} | AC_t^e}}$ . This means that the probability of  $T_{\text{ref}}$  is independent from how much more likely it is to reach  $T_{\text{ref}}$  than  $T_{\text{ext}}$  within the Boosted Ensemble.
- 2. the indicator variables  $(\mathbb{1}(T_h^m \ge T_{\text{ext}}))_{1 \le m \le N_h}$  and  $(\mathbb{1}(T_h^m \ge T_{\text{ref}}))_{1 \le m \le N_h}$  need to be independent.
- These assumptions will be discussed in the discussion section 4.3. Note that these assumptions do not require that  $T_{\text{ref}}$  and  $T_{\text{ext}}$  occur independently of one another, but rather explore the relationship between  $T_{\text{ref}}$  and  $T_{\text{ext}}$  within the boosted ensemble, and their link to  $T_{\text{ref}}$ .

The choice of an intermediate temperature threshold to create a link between the reference climate model simulation and the Boosted Ensemble has been studied, albeit under different conditions, in Finkel and O'Gorman (2024). However, beyond the higher complexity of the climate model used here for Ensemble Boosting compared to that of the Lorenz 96 simulator used in Finkel and O'Gorman (2024), the most important methodological difference lies in the process of generating perturbed offspring. While Finkel and O'Gorman (2024) boost one parent with a batch of 1 repeatedly until reaching the desired extremes, calculating conditional probabilities at each step, we perturb larger batches for ranges of lead times (as detailed above) only once. The former approach would lead to a more targeted result, with fewer unused simulations. However, it would also take substantially more time to perform all computations, since there would be no possibility of parallelization: given our current computational capacity (simulating approximately 1.1 climate model years in 24 node hours, with a capacity of up to 4 nodes), running the same 12 000 simulations generated for this study with this approach would take approximately 1.5 node years. Running large batches allows us to complete such setups within a few weeks. There is thus a trade-off between minimizing compute time using very effective sampling vs. minimizing wall-clock time by running more parallel simulations.

# 225 2.2 Theoretical comparison to the naive estimator



In order for the boosting estimator to be useful beyond finding return periods for temperatures that cannot be found in the reference climate model simulation, it needs to have lower errors than the naive estimator. In Appendix Section A, the variance and relative error of the boosting estimator are calculated, and can be shown to depend on the number of simulation years in the reference climate model simulation N, the total number of simulations in the Boosted Ensemble  $N_b$ , and the three probability terms  $\mathbb{P}(T \geq T_{\text{ref}})$ ,  $\mathbb{P}(T \geq T_{\text{ext}} \mid AC_t^{\epsilon})$  and  $\mathbb{P}(T \geq T_{\text{ref}} \mid AC_t^{\epsilon})$ .

Therefore, in order to calculate the theoretical relative error of estimating  $\mathbb{P}(T \geq T_{\mathrm{ext}})$ , we need to estimate the above unknowns. While the first three terms, N,  $N_b$  and  $\mathbb{P}(T \geq T_{\mathrm{ref}})$ , are parameters that can be set by the experimenter,  $\mathbb{P}(T \geq T_{\mathrm{ext}} \mid \mathrm{AC}^{\epsilon}_t)$  and  $\mathbb{P}(T \geq T_{\mathrm{ref}} \mid \mathrm{AC}^{\epsilon}_t)$  can only be calculated once the boosting experiment is performed, and will have to

Figure 3. Theoretical relative error of the boosting estimator with  $N=50\cdot 100$  days and (solid orange line)  $N_b=100\cdot 21$  days, (dashed orange line)  $N_b=500\cdot 21$  days and (dotted orange line)  $N_b=3000\cdot 21$  days. Theoretical relative errors computed for (a)  $\hat{p}_{T\geq T_{\rm ref}|AC_t^\epsilon}=0.75$ , and (b)  $\hat{p}_{T\geq T_{\rm ref}|AC_t^\epsilon}=0.3$ . Since a boosted simulation is less computationally expensive to run (see text), the naive estimator with a equivalent computational resource for each configuration of the boosting estimator is represented by a blue solid, dashed and dotted line, respectively. The relative error of the naive estimator with (solid line)  $N=4000\cdot 100$  days and (dashed line)  $N=50\cdot 100$  days are shown in black.

be approximated based on empirical evidence. Furthermore,  $T_{\rm ext} \geq T_{\rm ref}$  and  $T_{\rm ext}$  could exceed any  $T_b^m$  from the Boosted Ensemble. Therefore,  $\mathbb{P}(T \geq T_{\rm ext} \mid \mathrm{AC}_t^\epsilon) \in [0, \mathbb{P}(T \geq T_{\rm ref} \mid \mathrm{AC}_t^\epsilon)]$ . For illustration purposes, we choose values of 0.75 and 0.3 for the estimation of  $\mathbb{P}(T \geq T_{\rm ref} \mid \mathrm{AC}_t^\epsilon)$ , which amounts to cases where either 75% or 30% of the boosted simulations exceed  $T_{\rm ref}$ . These values were chosen because they correspond to the typical value we find in practice (see Results section 3.2).


The evolution of the theoretical relative error of the boosting estimator with  $\mathbb{P}(T \geq T_{\mathrm{ext}})$ , for different values of the above mentioned parameters, can be seen in Figure 3. Here, three different configurations of the boosting estimator are compared to the errors of the naive estimator. Since we need to run a sample of non-boosted parents before generating boosted samples, the total cost of the boosted simulations takes this into account  $(N+N_b=50\cdot 100+500\cdot 21\text{ days})$ , for a parent ensemble of size 50, where each simulation is run for 100 days, and a boosted ensemble of size 500, where each simulation is run for 21 days). To directly compare these results with the naive estimator, we generate a non-boosted sample where N is equivalent, in terms of computational resources, to those of each boosting configuration. For example,  $N=105\cdot 100$  days is equivalent to  $N+N_b=50\cdot 100+500\cdot 21$  days, since a boosted simulation only needs to be run for 21 days, while the reference climate model simulation needs approximately 100 days to generate a full summer.

An important result of this computation is that the relative errors of the boosting estimators are constrained by that of  $\mathbb{P}(T \geq T_{\mathrm{ref}})$ : any estimate of  $\mathbb{P}(T \geq T_{\mathrm{ext}})$  will have errors equal to or higher than  $\mathbb{P}(T \geq T_{\mathrm{ref}})$ . An accurate estimate of  $\mathbb{P}(T \geq T_{\mathrm{ref}})$  is therefore necessary for robust results. This advocates for using either a longer test slice as the reference climate model simulation or a less extreme (and thus better sampled)  $T_{\mathrm{ref}}$ .

Nonetheless, the boosting estimator can reduce the relative error compared to that of the naive estimator. Firstly, in all boosting configurations, relative errors are smaller than for the naive estimator with  $N=50\cdot 100$  days. Secondly, each boosting estimator reduces errors compared to its equivalently expensive naive estimator for probabilities under a certain threshold value of  $\hat{p}_{T\geq T_{\rm ext}}$ , which depends on  $N+N_b$ . Finally, the boosting estimator with  $N=50\cdot 100, N_b=3000\cdot 21$  days reduces errors compared to the much more expensive naive estimator with  $N=4000\cdot 100$  days in both Figure 3a and b, for a low enough probability  $\hat{p}_{T>T_{\rm ext}}$ .

Finally, Figure 3 also shows that for both  $N_b$  and  $\mathbb{P}(T \geq T_{\mathrm{ref}} \mid \mathrm{AC}_t^{\epsilon})$ , the relative error decreases as these parameters increase. There is therefore no optimal number of boosted simulations  $N_b$ : the larger the better. Finding the optimal  $\mathbb{P}(T \geq T_{\mathrm{ref}} \mid \mathrm{AC}_t^{\epsilon})$  is however less trivial because it creates trade-offs. These will be further discussed in Results section 3.2.

#### 260 2.3 Validation with an Ornstein-Uhlenbeck process




Before estimating return periods in a fully-coupled climate model, a validation and exploration of parameter settings is performed with a simple Ornstein-Uhlenbeck process, also called red-noise process. The objective of this validation, beyond checking that the estimator yields correct results, is to better quantify the uncertainties surrounding the boosting estimator, by separating the effect of parameter settings and random variability. Due to the lack of computational constraints in running the Ornstein-Uhlenbeck process, larger samples of simulations can be generated, which strengthens confidence in the conclusions drawn.

The evolution of this process obeys the following stochastic equation:

$$dX(t) = -\alpha X dt + \sigma dW(t), \tag{11}$$

where the statistical parameters  $\alpha$ ,  $\sigma$  are equal to 1 here, and  $dW(t) \sim \mathcal{N}(0,dt)$  is a Wiener process. It is composed of two terms: the first one,  $-\alpha X dt$ , models a deterministic drift towards 0, while the second one,  $\sigma dW(t)$ , generates randomness that simulates natural variability. This creates the characteristic mean-reverting evolution of the Ornstein-Uhlenbeck process.

We simulate  $10^6$  parallel simulations of this process that span  $100 \cdot \tau$ , with  $\tau = \frac{1}{\alpha}$  the de-correlation time of the process. This assures independence from each simulation's initial value, which is sampled from the stationary distribution of the Orstein-Uhlenbeck process. The set of maximum values, or events, found for each of these parallel simulations are calculated to act as a ground truth. From this ground truth, we sample N = 1000 events from which we select the parent ensemble. The boosted ensembles with varying  $N_{\text{parent}}$  and  $N_{\text{batch}}$  are subsequently generated. Since the dynamics of spread in a red-noise process is not directly comparable to that of a fully-coupled climate model, we keep  $N_{\text{lead}} = 1$ , and perturb at a lead time  $t = 0.2 \cdot \tau$ . Other lead times were tried, with no substantial change in results (not shown). Each simulation is then run until it is no longer

Return period estimation for boosted simulations from an Ornstein-Uhlenbeck process

Figure 4. Estimating return periods from boosted simulations in an Ornstein-Uhlenbeck process. (a) Evolution in time of one parent (blue) and its boosted simulations (orange). The mean  $\pm$  3 standard deviations of the boosted simulations is shaded in orange. Additionally, the theoretical mean (dashed line) and mean  $\pm$  3 standard deviations (solid line), as derived in Appendix Section B, are shown in black. The time at which the maximum theoretical boosted intensity is reached is highlighted by black dotted lines.

Estimated return periods from boosted simulations for (b)  $N_{\rm parent}=10$ , and  $N_{\rm batch}=10$ , (c)  $N_{\rm parent}=100$ , and  $N_{\rm batch}=10$ , (d)  $N_{\rm parent}=10$ , and  $N_{\rm batch}=100$ , and (e)  $N_{\rm parent}=100$ , and  $N_{\rm batch}=100$  are shown in orange: the median value, sampled from 1000 boosting experiments, is shown with a solid line, while the 95% confidence interval is delineated by dotted lines. In each panel, the ground truth is shown in black, while the 1000 repeated samples of the simulations used to select the parent ensemble are shown in blue (the median value as a solid line, and the 95% confidence interval is shaded).

correlated to its parent (time  $\simeq \tau$ ), and its maximum value is calculated. This process is repeated 1000 times to estimate uncertainties.






The evolution in time of one parent simulation, along with the mean  $\pm$  3 standard deviations of 1000 boosted simulations is shown in Figure 4a. Since the Ornstein-Uhlenbeck process employed here is fully stochastic, the typical spread of boosted simulations from their parent does indeed differ from the evolution of boosted simulations in a fully-coupled climate model as seen in Figure 1. However, the spread between members, illustrated by the range of the mean  $\pm$  3 standard deviations confidence interval, grows and saturates faster than the memory of the parent intensity, shown by the median of the boosted simulations, fades and reverts back to zero. This opens a window of time when one can sample events more intense as the parent  $x_0$ . As derived theoretically in Appendix Section B, for the Orstein-Uhlenbeck process, if one wants to sample anomalies of size  $k\sigma$ , with k > 0 being a function of the number of members in the boosted ensemble, then the time where the maxima are expected to happen (i.e. an approximation of the "optimal lead time") is:

290 
$$t^* = \frac{1}{2\alpha} \ln(1 + k^2 \frac{\sigma^2 / 2\alpha}{x_0^2}).$$
 (12)

As a sanity check, this equation can be applied to our climate model setup (see Subsection 2.4). When considering the evolution of temperature in summer, one would typically have  $1/\alpha \simeq 10$  d (the de-correlation time scale in the atmosphere),  $\sigma/\sqrt{2\alpha} \simeq 3.5$  K (the climatological standard deviation of summer 5-day rolling average of daily maximum temperature anomalies over the region of interest),  $x_0 \simeq 10$  K (the typical value of  $T_{\rm ref}$ ), and  $k \simeq 3$  (the maximum number of standard deviations one can expect to sample using  $N_b \simeq 100$  to 1000 boosted members per parent). This leads to  $t^* \simeq 15$  d, with expected maximum event magnitudes around 4 climatological standard deviations, i.e. temperature anomalies of 14 K.

Figure 4b-e shows return period estimates in the boosted ensemble, calculated with the boosting estimator for combinations of  $N_{\rm parent}=10,100$  and  $N_{\rm batch}=10,100$ . In all panels, the return period of the ground truth set, estimated with Equation 1, (in black) and sample sets from which we select each parent ensemble (in blue) are shown for comparison. Firstly, we see that the boosting estimator is indeed unbiased – all median return period estimates using the boosting estimator follow the ground truth, regardless of the configurations of  $N_{\rm parent}$  and  $N_{\rm batch}$ . Secondly, in all configurations of  $N_{\rm parent}$  and  $N_{\rm batch}$ , higher return periods, up to more than an order of magnitude, are estimated in the boosted ensemble than in the parent ensemble.

Additionally, while median return period estimates of the sample sets from which we select the parent ensemble also follow the ground truth (as expected, since both are estimated using Equation 1), the uncertainty of the return periods estimated from the boosted simulations seem to increase more slowly than for the non-boosted simulations in the sample set. Indeed, the 95% confidence interval around the return periods of the boosted simulations are narrower than those of the sample set for return periods above  $10^2$ . As predicted from the theoretical relative error calculation (see Section 2.2), this reduction of uncertainty is more pronounced for a larger  $N_{\text{batch}}$  (Panels  $\mathbf{d}$ , $\mathbf{e}$ ) or  $N_{\text{parent}}$  (Panels  $\mathbf{c}$ , $\mathbf{e}$ ), with the smallest uncertainty range found in Panel  $\mathbf{e}$ , where the total computational cost is largest (10 000 simulations). This highlights that while sparse sampling of either the parent or the boosted ensemble can increase uncertainty and relative error with respect to the ground truth, it does not bias the return period estimation.

Finally, comparing Panels  $\bf c$  and  $\bf d$  allows for evaluating how  $N_{\rm parent}$  and  $N_{\rm batch}$  affect the boosted simulations and their subsequent return period estimation. At the same computational cost as Panel  $\bf c$ , Panel  $\bf d$  presents boosted simulations with higher return levels and similar, or at times smaller, levels of uncertainty. This is because  $\mathbb{P}(T \geq T_{\rm ref})$  here is much higher (0.01 compared to 0.1 for Panel  $\bf c$ ), thus increasing the chances of events in the boosted ensemble with larger return periods.

By exploring different configurations of the relevant boosting parameters  $N_{\rm parent}$  and  $N_{\rm batch}$ , this evaluation thus strengthens confidence in estimating return periods using the boosting estimator. This confidence is underscored by a more extensive uncertainty sampling of both the parent and boosted ensemble than what would be possible in a fully-coupled climate model. However, it is important to note that the Ornstein-Uhlenbeck process is one-dimensional, while a state-of-the art climate model has millions of degrees of freedom. Therefore, the transferability of results found here are limited, in particular for the size of the parent ensemble necessary to sample  $AC_t^{\epsilon}$  well.

# 2.4 Experimental setup of Ensemble Boosting in a fully-coupled climate model








To show how the theoretical properties of the boosting estimator apply in a climate model context, beyond a simple Ornstein-Uhlenbeck process, we generate simulations with the state-of-the-art fully coupled Community Earth System Model 2.1.2 (Danabasoglu et al., 2020). We seek to estimate the return levels of the yearly maximum of 5-day rolling average of daily maximum temperature anomalies (TXx5d) spatially averaged over the region of the PNW heatwave (45–52°N, 119–123°W), corresponding to the region used by Fischer et al. (2023).

First, we test our boosting estimator in a pre-industrial setting. A 4000-year long pre-industrial control run is used to act as a control period, while two 50-year time slices of this simulation, test slice 1 (1801–1850) and test slice 2 (1851–1900), act as our reference climate model simulations. While these ranges are adjacent to one another, the starting point of the total time range is selected randomly within the control run. Additionally, since the time is only referenced as time since the start of the simulation, the years do not bear any meaning or relation to real-world weather at that time. Two separate test slices are used to increase the robustness of the results. Since they only span 50 years, extremes (i.e. events with a return period longer than 50 years) will be scarce or absent and their return period estimates highly uncertain. The length of the time slice is selected to reflect typical timescales of available historical records. While this can serve as a comparison to historical extreme event attribution studies, this time scale might present highly uncertain estimations of  $T_{\rm ref}$ , in particular due to the limited sample and the long term temperature variability effects. All events in the Parent Ensemble, selected according to the boosting algorithm above, are boosted according to the algorithm detailed above, and their return periods are calculated using Equation 10. This is done independently for both test slices. The results can then be directly compared against the more robust return periods found in the control period. Additionally, we fit stationary GEV distributions using both the maxima of the control period and of the test slices. This is done to give an estimate of return periods beyond these observational records with a standard method from Extreme Value Theory. A bootstrapping approach is used to calculate 95% confidence intervals, fitting GEV distributions to 1000 resamples drawn randomly from the test slices and the control period.

In each test slice, the five most extreme years are selected to form the Parent Ensemble, which corresponds to taking  $T_{\rm ref}$  such that  $\mathbb{P}(T \geq T_{\rm ref}) = 0.1$ . This value is chosen to balance rareness in the Parent Ensemble and the ability to robustly estimate

 $\mathbb{P}(T \geq T_{\mathrm{ref}})$  in the test slice. Changing the threshold of selection is equivalent to either changing  $T_{\mathrm{ref}}$  and keeping the sample size equal (which means changing the number of parents) or keeping  $T_{\mathrm{ref}}$  but changing the sample size. The former will be evaluated in the discussion section, and the latter in the following subsection. In order to sample the uncertainty of  $\mathbb{P}(T \geq T_{\mathrm{ref}})$ , this value is estimated by bootstrapping the relevant test slice sample. These are then subsequently boosted for all lead times in the range of -7 to -18 days, with batches of 100 simulations per lead time and per parent event. This leads to a total of 6000 boosted simulations for each test slice. The maximum 5-day daily maximum temperature (Tx5d) anomaly over the 21 days following perturbation is assessed, so that the block maximum approach stays the same as for the Parent Ensemble. The length of 21 days is taken to correspond roughly to the saturation time of the boosting algorithm, i.e. when the growing divergence between boosted simulations have saturated to that of the reference climate model simulation. This is in line with the time scale of classical weather predictability of around 10 days (Krishnamurthy, 2019). The exact length is determined through empirical trial and error, and will be discussed in Results section 3.2.

In order to directly compare return periods calculated using the boosting estimator, the TXx5d need to reasonably fulfill the domain convergence conditions of GEV theory. While 21 days is not as long as the approximately 100 days of a full summer, we postulate that the boosted simulations comply reasonably well with EVT requirements for the following reasons: Firstly, since  $T_{\rm ref}$  is already a rare summer maximum, the boosted simulations that exceed it are likely to also be the summer maximum. Indeed, for the simulations that do exceed  $T_{\rm ref}$ , only 4-5% have a higher temperature in the following 60 days. Secondly, the parent heatwaves do not all occur in the beginning of the summer. Since the boosted simulation shares the trajectory of its parent until it is perturbed, the boosted simulations are generally longer than just 21 days.

A bootstrapping procedure is also performed to calculate a confidence interval around the estimated return period from the boosting estimator. However, since the number of boosted simulations where  $T_{\rm ext} \geq T_{\rm ref}$  is not always the same in random samples of the Boosted Ensemble (since some boosted simulations do not exceed  $T_{\rm ref}$ ), the bootstrapping proceeds as follows: for each  $T_{\rm ext} \geq T_{\rm ref}$  in the Boosted Ensemble, 1000 random samples of the Boosted Ensemble are generated to calculate  $\mathbb{P}(T \geq T_{\rm ref} \mid AC_t^{\epsilon})$  and  $\mathbb{P}(T \geq T_{\rm ext} \mid AC_t^{\epsilon})$ , and thus estimate  $\mathbb{P}(T \geq T_{\rm ext})$ . Note that since we do not change the experimental sample when calculating the confidence interval (e.g. selecting and boosting a different parent ensemble), this confidence interval may underestimate certain sources of uncertainty such as slow modes of variability of the climate system.

Second, we boost simulations from the CESM2.1.2 climate model in present-like conditions (2005-2035, SSP3-7.0 after 2015) to estimate the return period of the record-shattering 2021 Pacific North-West heatwave. A 30-member Large Ensemble is used as the reference climate model simulation, leading to take  $N=30\cdot31=930$  years. From this initial simulation, 7 among the 13 most extreme events are selected to form the Parent Ensemble, which is equivalent to take  $\mathbb{P}(T\geq T_{\rm ref})=\frac{13}{930}=0.014$ .

Due to the presence of anthropogenic forcings in the reference climate model simulation from 2005 to 2035, the non-stationarity in the underlying statistical distribution needs to be accounted for. This is done by linearly de-trending the TXx5d time series: for each year and each member of the Large Ensemble, a 100-member Large Ensemble of the same model (Rodgers et al., 2021) is used to produce a day-of-year mean across members and a three-year window (the year in question, and one year before and after) around each year. The three-year window was chosen to create a larger sample of similar years, without adding the climate change signal present over longer time scales. While simple de-trending could produce artifacts in the

distributional tail, the forcings in this historical and near-future sample is small enough for the correction to be considered a reasonable approximation.

The 100-member Large Ensemble spans the same time range (31 years) and is also corrected for non-stationarity in the same way. However, this publicly available data set is not locally bit-by-bit reproducible, which is necessary to generate boosted simulations. It will therefore act as a separate, larger data set that can provide return periods that are more precise than those of the 30-member Large Ensemble, since here,  $N = 100 \cdot 31 = 3100$  years.

Note that the 7 selected events for present-like conditions are not, contrary to the pre-industrial control case, those with the absolute highest TXx5d, but rather 7 among the top 13 events. This is because the correction for non-stationarity was performed after the selection process, which was determined in a previous study (Fischer et al., 2023). In the context of the boosting estimator, this simply corresponds to a more sparse sampling of  $AC_t^{\epsilon}$  than what is maximally possible with the climate model simulation at hand: one could imagine a reference climate model simulation with only these 7 events present.

To calculate the TXx5d of the 2021 PNW heatwave, a detrended time series of the ERA5 reanalysis data set of the ECMWF (Hersbach et al. (2020)), regridded to fit the CESM2.1.2 model grid, is used. The return period of this return level is then calculated using the boosting estimator, and, for comparison, the GEV fits of the 30- and 100-member Large Ensembles. A median (percentile) return period estimate is written as  $\infty$  when the median (percentile) probability is 0.

## 3 Results






## 3.1 PI-control runs

Figure 5 shows return period estimates for TXx5d in the Boosted Ensemble, calculated with the boosting estimator, for test slice 1 (Panels  $\mathbf{a}$ , $\mathbf{b}$ ) and test slice 2 (Panels  $\mathbf{c}$ , $\mathbf{d}$ ). For comparison, return period estimates calculated with the naive estimator are shown for the control period, and a GEV fit of the test slice itself provides an extrapolation outside the fitting period. In order to increase diversity and robustness among the boosted simulations, results are calculated by pooling together perturbation lead times from -18 to -13 days in Panels  $\mathbf{a}$ , $\mathbf{c}$ , and -12 to -7 days in Panels  $\mathbf{b}$ , $\mathbf{d}$ . This categorization remains somewhat arbitrary, however; results calculated lead time by lead time for test slices 1 and 2 can therefore be seen in Appendix Figures A2 and A3, respectively.

In Figure 5, the results found using the boosting estimator stand in stark contrast to those of the non-boosted test slices. First, the maximum TXx5d sampled through boosting substantially exceed those of the test slices, by up to  $2.9^{\circ}$ C, and even reaches the maximum TXx5d of the control period (Figure 5b, d). Second, return period estimates of the simulations in the two test slices deviate more from the control period than those of the Boosted Ensemble, in particular for Test slice 1. This follows the theoretical results presented in Figure 3: for  $N_b = 3000$  years, any configuration of the boosting estimator should lead to return period estimates with less relative error than for the naive estimator with N = 50 years. The GEV distribution fitted to the test slices is also error-prone; for return periods up to  $\frac{1}{\mathbb{P}(T \geq T_{ref})}$ , estimates calculated with the boosting estimator in all configurations (Panels a, b, c, d) follow the control period better than the median GEV fits of the test slices. It is also worth noting that the confidence interval of the GEV fit of both test slices is much wider than that of the boosting estimator; in

Figure 5. Return periods of TXx5d in the PNW region estimated with the boosting estimator under stationary climate conditions. Estimated return periods with the boosting algorithm for parent events selected  $(\mathbf{a},\mathbf{b})$  from test slice 1 (1801-1850) and  $(\mathbf{c},\mathbf{d})$  from test slice 2 (1851-1900) of the pre-industrial control simulation are shown in orange. Perturbation lead times are pooled together from  $(\mathbf{a},\mathbf{c})-18$  to -13 days and  $(\mathbf{b},\mathbf{d})-12$  to -7 days. In each panel, the TXx5d of the 4000 years of the pre-industrial control simulation are shown in black. The TXx5d of the two 50-year test slices are shown in blue. The five selected parent events in each test slice are highlighted in diamonds. The horizontal dashed orange line represents the reference TXx5d  $T_{\text{ref}}$  in each test slice. For the return periods of the control and test slice simulations, a GEV law is fitted and the estimated return period is shown (solid line) with a bootstrap 95% confidence interval (shaded). For the boosted simulations, the shaded area shows the bootstrap 95% confidence interval (see Methods).

particular, the upper bound of the confidence interval of test slice 2 is unbounded. These deviations from the control period are to be expected given the short climate model record (N = 50 years) – both Zeder et al. (2023) and Noyelle et al. (2024a) have shown that GEV fits from short records are prone to systematic biases.

Furthermore, Figure 5a shows that return period estimates, calculated with the boosting estimator for lead times from -18 to -13 days in test slice 1, follow the estimate using the control period for all  $T_{\rm ext}$  present in this sample: both median boosting estimates and the confidence interval overlap with the control period remarkably well. Results from test slice 2, for lead times from -18 to -13 days (Figure 5c), indicate that estimates using the boosting estimator also can deviate from the control period, although they remain within an order of magnitude compared to the control period estimate. A reason for this deviation might be that the test slices span only 50 years, and only 5 parent events were selected for boosting. Therefore, it is possible that the Parent Ensemble does not sample  $AC_t^{\epsilon}$  sufficiently well, and that the return period estimates calculated are actually conditional on a particular feature of long term natural variability present in test slice 2 only. In Appendix Figure A4, the return periods calculated by combining the two test slices into one 100-year long time series are shown. Here, the return periods again follow the control period very well.

For shorter lead times, between -12 and -7 days, return period estimates deviate from the control period confidence interval already at around 30–40 years for simulations from both test slices (Figure 5b,d). This is likely because the short lead time leads to simulations that are too constrained by the maximum TXx5d of the parent events, since they do not have enough time to deviate enough from their parents. In Appendix Figures A2 and A3, it can be seen that this deviation is gradual, that appears around -13 days for Test slice 1, although for Test slice 2, the picture is less clear — some deviation is present already at -18 days. In particular, we observe "step-like" behavior of the boosted return periods around the values of the parent events for shorter lead times (Appendix Figure A2j,k,l), meaning that the TXx5d of the boosted simulations are very similar to that of their parent, forming return levels that resemble a step function. If the boosted simulations are too close to their parents, this additionally means that there is less independence between boosted simulations, thus breaking one of the assumptions made in the methods to show the unbiasedness of the boosting estimator. This could also contribute to why we have such a discrepancy for short lead times.

It is worth noting that the increase in estimation uncertainty, illustrated by a widening of the confidence interval as return periods increase, shows that the most extreme events are estimated less precisely. Additionally, the confidence intervals should be interpreted somewhat cautiously, since they are computed through bootstrapping. This means they are fundamentally limited by the sample at hand, and not representative of all uncertainty factors such as potential multi-decadal variability beyond the time scales sampled here.

#### 3.2 Sensitivity tests: theoretical assumptions and parameter choices







The difference in results between longer and shorter lead times, seen in Figure 5, as well as the differences in relative error depending on  $N_b$  and  $\mathbb{P}(T \ge T_{\text{ref}} \mid AC_t^{\epsilon})$  seen in Figure 3, show that the parameter choices and assumptions made can influence the quality of the estimation obtained. This warrants a deeper analysis of these choices and assumptions.

First, the effects of the lead time chosen when boosting are assessed beyond the separation of short and long lead times in Figure 5. Figure 6 shows the TXx5d of the boosted simulations of the parent events separately, as a function of lead time.

Although there are variations between simulations stemming from different parent events, certain patterns are visible. For all panels, the spread between boosted simulations shrinks as the lead time grows shorter, centering around the parent peak. This

Figure 6. TXx5d of boosted simulations as a function of lead time and parent event. Boosted simulations from parents 1-5 in (a–e) test slice 1 and (f–j) test slice 2. For each parent event a batch of  $N_{\text{batch}} = 100$  boosted simulations are generated at every lead time. The median (orange solid line),  $5^{\text{th}}$  to  $95^{\text{th}}$  percentile range (orange shaded area), and maximum TXx5d (orange plus-sign) are shown. TXx5d for the parent event is represented by a blue horizontal line.

corresponds to boosted simulations being more and more constrained by the dynamics of their parents. Conversely, for longer lead times, the median tends to decrease and the spread tends to grow. In other words, simulations diverge from the parent (and each other) so that the memory of the parent heatwave fades and simulations revert back to the underlying climatology. There are exceptions to this general picture, however, in particular for parent events from Test slice 2. Here, simulations from parent events 1, 2 and 5 seem to show less sensitivity to lead times, with relatively consistent levels of spread and medians and visibly less spread than simulations from other parent events at long lead times.



It is also worth noting that from one lead time to another, both median, spread and maximum values can vary substantially. Additionally, the maximum TXx5d in a given panel is not always from the lead time with the highest median or  $95^{\rm th}$  percentile. This suggests that several lead times might be necessary to better sample the antecedent conditions  $AC_t^{\epsilon}$ .

The effect of the day at which perturbation is performed on the estimation of return periods using the boosting estimator is broken down in more detail in Figure 7. Panel **a** shows how simulations spread after perturbation: we see that for both test

Figure 7. Link between boosted simulations and perturbation time in (blue) test slice 1 and (orange) test slice 2. (a) Evolution of boosted simulation standard deviation between members with respect to the time elapsed since perturbation. Standard deviation between boosted simulations, divided by the standard deviation of the pre-industrial control simulation: (solid line) median spread across parent events and (shaded area) range of spread across parent events. (b) Frequency of events exceeding  $T_{\text{ref}}$  in the Boosted Ensemble ( $\hat{p}_{T \geq T_{\text{ref}}|AC_t^e}$ ) as a function of lead time. Error-bars denote the bootstrapped 95% confidence interval.

slice 1 and test slice 2, the spread between simulations initially grows exponentially, before slowing into a linear growth phase and finally saturating to the climatological spread between the simulations in the non-boosted ensemble of the control period. This type of growth is described in ergodic chaos theory, which shows that initially, boosted simulations diverge from their parent exponentially fast with a typical time scale around the inverse of the largest Lyapunov exponent of the system, and has been extensively studied for both climate and weather forecast models (e.g., Trevisan and Palatella, 2011; Vannitsem, 2017). Boosted simulations with short lead times are thus still in the exponential spread stage by the time the heatwave peaks, and will therefore be significantly constrained by the value of the parent event at that time. This means that the dynamics governing the heatwaves in the boosted simulations are structurally too similar to those of the parent event to produce a largely different TXx5d value.



Note that, given the results of Figure 7a, the empirical saturation time after boosting is set to 21 days (see Methods). This means that the maximum Tx5d of the boosted simulations are only assessed in this time range, because afterwards, the memory of the parent, and thus the antecedent conditions leading to a heatwave, are assumed to be lost.

Figure 7b, on the other hand, shows that  $\hat{p}_{T \geq T_{\text{ref}} | AC_t^{\epsilon}}$  increases as the lead time grows shorter. Since Figure 3 shows that a higher  $\hat{p}_{T \geq T_{\text{ref}} | AC_t^{\epsilon}}$  leads to smaller theoretical errors, shorter lead times should give fewer errors. In other words, a smaller  $\hat{p}_{T \geq T_{\text{ref}} | AC_t^{\epsilon}}$  indicates that there are fewer simulations that surpass  $T_{\text{ref}}$ , and thus less extreme events to find return periods for

Figure 8. Relationship between TXx5d of parent event (blue diamond) and boosted simulations (orange box plot) for (a) test slice 1 and (b) test slice 2. Simulations are pooled together from lead times between -18 and -13 days. Boxplot whiskers are drawn at 1.5 inter-quantile range.

overall. However, as previously mentioned, the longer the lead time, the more each simulation has had time to spread, and thus gain independence from both parents and siblings. Therefore, a trade-off between relative error minimization and sampling more intense events appears. Note that one cannot reliably estimate the spread as a function of lead time, since each boosted simulation may not exactly follow the theoretically predicted ergodic growth speed, in particular due to the extreme nature of the parent events. This is evidenced by the large variation of growth exhibited between boosted simulations from different parents (see Figure 7). Therefore we cannot explicitly state the boundary lead times of this trade-off, in particular since any effort to do so would be both location- and variable-specific (heatwaves in the PNW region).




The above-described effects do not all set in at the same time across parent events and test slices. Figure 7a shows that while the median spread overlaps well between Test slice 1 and 2 until around 18 days after perturbation, the saturation sets in earlier for Test slice 2. Figure 7b also shows some differences: while both test slices have the same trend, simulations from Test slice 1 seem to have higher  $\hat{p}_{T \geq T_{\text{ref}}|AC_t^e}$  than for simulations from Test slice 2 until a lead time of around -10 days. Pooling the simulations perturbed at different lead times, like in Figure 5, may therefore provide more robust results. This has the added benefit of adding more independence between boosted simulations.

Finally, Figure 8 shows that there is only a weak relationship between the TXx5d of the parent event and that of its boosted simulations. Indeed, the Spearman correlation between the 90<sup>th</sup> percentile of the boosted simulations and the 10 parent events, shown in Appendix Figure A5, is only 0.3. Furthermore, Figure 8 shows that the boosted simulations present a remarkable

Figure 9. Return periods of TXx5d in the PNW region estimated with the boosting estimator corrected for non-stationary (climate change) conditions for a lead time range of -18 to -13 days. Estimated return periods for boosted simulations from (a) the entire Parent Ensemble and (b) the entire Parent Ensemble except the most extreme parent (E1) are shown in orange. The TXx5d of the (black dots) 100-member and (blue dots) 30-member Large Ensemble, with a fitted GEV law and shown with the median (solid line) and a bootstrapped 95% confidence interval (shaded area). The parent events in the Parent Ensemble are shown with blue diamonds, and the most extreme parent E1, and the most extreme event in the 100-member Large Ensemble  $\tilde{E}1$  are highlighted with stars. The orange horizontal line denotes the reference TXx5d  $T_{\rm ref}$ , while the black horizontal line denotes the 2021 ERA5 TXx5d.

variation, not only in the distance from their parent, but also in their median and spread. It is, however, important to note that the boosted simulations generated for this study only belong to a limited number of parent events ( $N_{\text{parent}} = 5$  for each test slice), thus preventing us from concluding on a wider basis. Nevertheless, this suggests that a larger Parent Ensemble, i.e. a wider variety of parent events, would not necessarily lead to less extreme boosted simulations, at least not for parents above a certain threshold of intensity.

## 3.3 Estimating a return period for the Pacific North-West heatwave of 2021


As a further application and a case study, we now use the boosting estimator to estimate a return period, based on the CESM2 climate model, for the record-shattering PNW heatwave of 2021. To do this, we employ a present-climate 30-member Large Ensemble corrected for non-stationarity (see Method Section 2.4) as a reference simulation, from where boosted simulations are generated. The median return period of the 2021 PNW heatwave is calculated using the boosting estimator, and found to be 2500 years with a 95% confidence interval of 2000 to 4000 years. In comparison, the GEV fit of the reference 30-member

Large Ensemble and the 100-member Large Ensemble give higher median return period estimates, of  $10^6$  [10 000,  $\infty$ ] years and 15 000 [5000,  $10^7$ ] years, respectively.







Figure 9a shows all return periods calculated with the boosting estimator for boosted simulations with  $T_{\rm ext} \geq T_{\rm ref}$ . Lead times are chosen to be [-18,-13], corresponding to the lead times that did not show significant constraints by parent events in the pre-industrial control experiment. A lead time by lead time breakdown of return periods is nevertheless shown in Appendix Figure A6. In Figure 9a, we see that the median return period estimates deviate from those of the 30- and 100-member Large Ensembles and their GEV fits for return periods between around 200 and 10 000 years, although this deviation is always within an order of magnitude error of the upper bounds of the GEV fit of both the 30- and the 100-member Large Ensemble. This deviation is more present for shorter lead times, not included in Figure 9, but visible in Appendix Figure A6.

One hypothesis that could explain this deviation is that the boosted simulations from the most extreme parent event, hereafter denoted by E1, are biased due to the intensity of the event. Indeed, E1 stands in contrast to the rest of the events in the Parent Ensemble — with a TXx5d of 14.7°C, it is 1.6°C warmer than the second most extreme parent event E2. Remarkably, the 100-member Large Ensemble also presents such an event,  $\tilde{E}1$ , with TXx5d of 15.88°C, that is 2.0°C warmer than  $\tilde{E}2$ .

To test this hypothesis,  $9\mathbf{b}$  shows return periods calculated excluding boosted simulations from E1. These return period estimates follow the  $95^{\text{th}}$  percentile of the uncertainty range of the 100-member Large Ensemble GEV fit. Additionally, this effect is unique to the removal of boosted simulations from E1: when removing any other parent event, the results look largely the same (see Appendix Figure A7).

We also see that removing E1 has a strong impact on the GEV distribution fit of the 30-member Large Ensemble data: in Figure 9a, the confidence intervals of black and blue fits overlap significantly, while in 9b, they are almost disjoint. This highlights the previously stated uncertainty of the bootstrapped confidence interval of the 30-member Large Ensemble, and suggests that the presence of E1 in the Parent Ensemble largely affects the naive return period estimation. Indeed, when removing E1 from the Parent Ensemble, the median return period estimate is infinite, with a confidence interval of  $10^8$  to  $\infty$  years. Removing E1 when using the boosting estimator, on the other hand, leads to a median estimate of 3500 years with a 95% confidence interval of 2500 to 7000 years, which is within 1000 years of the original estimate.

In other words, had E1 not occurred in the reference simulation, the 2021 PNW heatwave and indeed E1 itself would be judged impossible with only the Large Ensemble data. However, the boosting estimator can provide return periods for extreme events like the 2021 PNW heatwave that depend less on E1, and that, importantly, stay finite. We can thus see that in addition to providing more robust return period estimates for extreme events, Ensemble Boosting also demonstrates that both E1 and the 2021 PNW heatwave events are physically possible, according to the climatology of this model.

To gain a picture of the atmospheric dynamics associated with E1 and the other most intense heatwaves in the 30-member Large Ensemble, the Tx5d anomaly and Z500 contour lines are plotted on the day of each heatwave peak in Figure 10. E1 distinguishes itself from the other top 13 events, and from the 2021 PNW heatwave by the presence of a more distinct cyclonic anomaly in the Z500 field over the Eastern Pacific, flanking the region of interest. Furthermore, the blocking high is centered around the PNW region for both E1 and the 2021 PNW heatwave, while the center of the blocking high shown in the composites is located further South. However, the E1 blocking high is more oblong and rotated due to the presence of the Pacific trough.

Figure 10. Tx5d anomaly [ $^{\circ}$ C] and Z500 [m] contour lines on the peak day of each heatwave, for (a) the most extreme event in the 30-member Large Ensemble E1, (b) ERA5 data of the PNW heatwave on 2021-06-28, and composites of (c) E2-3 and (d) E2-13, both from the 30-member Large Ensemble. The black box indicates the region of interest. The anomaly calculations are corrected for non-stationarity (see Methods section 2.4.

While the composite maps of E2-3 and E2-13 look distinct from that of E1, individual events within this selection could look different to their mean. Therefore, all 13 events were plotted individually in Appendix Figure A8. While there are events that show a third Z500 trough next to the PNW region (events E4 and E11), these are either less distinct from the main troughs forming the block, or fail to create a significant blocking high around the PNW region.

Additionally, when analyzing the most extreme from the 100-member Large Ensemble  $\tilde{E}$ , we see that it presents a similar dynamical situation to that of E1 (see Appendix Figures A9 and A10) with a third Pacific trough that is not seen in the other top events of the 100-member Large Ensemble.

## 4 Discussion



## 4.1 Reduction of errors and computational costs of the boosting estimator compared to the naive estimator

The theoretical derivation of the boosting estimator variance has shown that it can reduce the relative error compared to equivalently expensive naive estimators – and even an estimator based on the full 4000-year control period – given the right configuration and an extreme enough event (see Figure 3 and method section 2.2).

The computational resource use of the boosting estimator compared to a naive estimator can be assessed in more detail. In order to be useful as an estimator, generating the boosted simulations necessary to reduce errors compared to the control period should also be less computationally costly. For this comparison, will use  $N_b = 3000$  years, where relative error is reduced for values under a certain  $\mathbb{P}(T \ge T_{\rm ext})$ . The following estimate is calculated to compare these costs: generating  $N_b = 3000$  boosted simulations of 21 days and N = 50 parent summers amounts to generating 68 000 days of climate model simulations if the length of a summer is approximated to be 100 days. On the other hand, the N = 4000 summers of the control period would require 400 000 days. The boosting estimator could thus work as a more efficient way of estimating return periods of very rare extremes, since it would use approximately 6 times less computational resources, which given our computational capacities could save almost 20 000 node hours, and yield less erroneous results for a specific extreme of interest.

## 4.2 Effect of the number of parent events on return period estimates of the boosting estimator







Since the relative error of boosting estimator will always be at least equal to that of  $\hat{p}_{T \geq T_{\rm ref}}$ , selecting more parent events reduces errors because of a more precise estimation of  $\hat{p}_{T \geq T_{\rm ref}}$ : more parent events lead to a lower  $T_{\rm ref}$ , since this is the threshold that needs to be reached by all parents. A lower  $T_{\rm ref}$ , in turn, leads to a higher  $\hat{p}_{T \geq T_{\rm ref}}$ . Since  $\hat{p}_{T \geq T_{\rm ref}}$  is estimated naively, its relative error decreases when  $\hat{p}_{T \geq T_{\rm ref}}$  is higher (see Equation 3). This could help explain the discrepancy between test slices 1 and 2, since  $T_{\rm ref}$  differ between the test slices (9.78 in test slice 1 and 9.39°C in test slice 2). This leads to a relative error of  $\mathbb{P}(T \geq T_{\rm ref})$  of 15.9%, 16.4%, respectively, compared to the control period. While these numbers cannot act as a ground truth, since control period estimate itself also has errors, this finding still indicates that the parent event selection in test slice 2 could underestimate  $T_{\rm ref}$ .

In a similar vein, more parent events also increase the sample size of antecedent conditions in  $AC_t^{\epsilon}$ , making it more likely that this set is sufficiently well represented. Conversely, a non-representative Parent Ensemble could lead to error-prone results. Results from test slice 1+2 (see Appendix Figure A4) corroborate this line of reasoning, since this larger sample follows the control period estimate well. However, the typicality argument (Galfi and Lucarini, 2021; Lucarini et al., 2023; Noyelle et al., 2024b) states that the more extreme an event is, the more dynamically similar it is expected to be compared to other extremes of that magnitude. It follows that the number of events necessary to sample reasonably well  $AC_t^{\epsilon}$  may actually be small.

Finally, Figure A5 shows that the correlation between a parent event's intensity and its 90<sup>th</sup> percentile simulations is 0.3, which implies that in our limited sample, there does not seem to be a strong relationship between parent maximum and intensity of boosted simulations. One could imagine that this hypothesis breaks down if the selection threshold is lowered to include much less extreme parent events, since the dynamics of such events would be distinct from those of a heatwave. In this sense, there may exist a form of bias-variance dilemma for the Boosting Estimator: selecting more parents decreases the variance but increases the bias, where here "bias" would be understood as the lack of intensity of extremes simulated. It would therefore be necessary to test whether this holds for a larger number of parent events, but it indicates that more parent events would not necessarily give less extreme boosted simulations. All in all, while the optimal number of parent events is not found here, a general recommendation to set a large enough number of parent events can be stated.

## 4.3 Effect of perturbation lead times on return period estimates of the boosting estimator







If the dynamics of extremes studied can be approximated by red-noise dynamics, we have shown that Equation 12 gives an order of magnitude for the typical lead time to be used (15 days in our case) and the expected intensity of the extremes sampled (4 standard deviations, or 14 K anomalies in our case). These approximations are remarkably close to the values found empirically with the CESM2 climate model. This illustrates that the mechanism outlined in the red-noise process — i.e. the idea that there is a window of time for sampling more extremes while the variance increases, but before the mean reverts back to the climatology — is a theoretical justification for the possibility to sample more extreme events with Ensemble Boosting.

Additionally, as discussed in Results section 3.2, the optimal lead time at which boosting is performed is found through a trade-off between error minimization and independence between boosted simulations. Here, we empirically solve the present trade-off by pooling lead times where boosted simulations are not substantially constrained by their parent. The assumption of independence between the theoretical probability  $\mathbb{P}(T \geq T_{\rm ref})$  and the ratio  $\frac{\mathbb{P}(T \geq T_{\rm ext} \mid AC_t^{\epsilon})}{\mathbb{P}(T \geq T_{\rm ref} \mid AC_t^{\epsilon})}$ , which is postulated in order to prove the unbiasedness of the boosting estimator, is another argument in favor of pooling boosted simulations from longer lead times. This rather unintuitive relationship can be broken down to the stricter question of independence between  $T_{\rm ref}$  and the boosted simulations: if exceeding  $T_{\rm ref}$  is independent of exceeding extreme values obtained through boosting,  $\mathbb{P}(T \geq T_{\rm ref})$  will be independent of both  $\mathbb{P}(T \geq T_{\rm ext} \mid AC_t^{\epsilon})$  and  $\mathbb{P}(T \geq T_{\rm ref} \mid AC_t^{\epsilon})$ . To test this assumption, one would need to vary  $T_{\rm ref}$  (corresponding to varying the size of  $N_{\rm parent}$ ) and see how the boosted TXx5d varies.

Figures 8 and A5 show that this independence is a valid assumption in our samples, since the relationship between parent TXx5d and the median, spread and  $90^{\rm th}$  percentile of boosted TXx5d is weak. Given this weak relationship, we do not expect a clear trend in the resulting boosted TXx5d as more (less extreme) parent events are included. However, these figures are plotted for lead times between -18 and -13 days. With a shorter lead time, the boosted simulations will be constrained by their parent, therefore inducing a stronger relationship between  $N_{\rm parent}$  (and thus  $T_{\rm ref}$ ) and the resulting boosted simulations. Thus, longer lead times, or at least lead times where the boosted simulations aren't substantially constrained by the TXx5d of their parent, is recommended.

## 4.4 Critically assessing the estimated return period of the 2021 PNW heatwave

While the boosting estimator is a promising tool for estimating return periods within climate models, it is important to first underline the large uncertainties attached to raw climate model output. Climate models remain an imperfect representation of the
full Earth system, and comparison with observational data should be performed with caution, at the risk of over-interpretation.

Nevertheless, temperature anomalies, or temperature anomalies divided by climatological standard deviation has been used in
several studies when comparing observational and climate model data for estimating return times for the 2021 PNW heatwaves
(Bartusek et al., 2022; Malinina and Gillett, 2024; McKinnon and Simpson, 2022).

In Results section 3.3, it has been shown that return period estimates deviate from the confidence interval of GEV fits for return periods from the 30- and 100-member Large Ensemble — although only within less than one order of magnitude. This

deviation disappears when removing boosted simulations from E1 (see Figure 9). Two hypotheses can be put forth to explain this discrepancy.

Given the known shortcomings of GEV distribution extrapolations to calculate return periods, it might be that the return periods are not biased by E1, but rather that the GEV distribution fit is both overconfident (only capturing uncertainties from bootstrapping the limited sample, leading to a smaller confidence interval) and systematically overestimating return periods due to limitations of the sample size at hand. Additionally, a non-representative sampling of  $AC_t^{\epsilon}$ , could explain the sensitivity to boosted simulations from E1, in particular since only 7 of the 13 most extreme events from the reference simulation were selected as parent events (see Methods).





Another hypothesis is that E1 is dynamically distinct to the other heatwaves studied, thus justifying the removal of the boosted simulations perturbed from E1 when estimating return periods. This could be done since the antecedent conditions of E1 would not be a representative sample of  $AC_t^{\epsilon}$  when estimating a return period for the 2021 PNW heatwave. A dynamical analysis of the heatwave peak day corroborates this – both E1 and  $\tilde{E}1$ , from the 100-member Large Ensemble, seem to distinguish themselves from other parent events and observational data for the 2021 heatwave (see Figures 10 and A9). This hints at a potential bimodality in the dynamics of extremes in the tail, and somewhat contrasts the typicality argument presented above, which postulates that the more extreme a heatwave is, the more dynamically similar it is to heatwaves of the same intensity. Yet, previous studies on the typicality of heatwaves also have indicated the possibility of such a tail bimodality (Noyelle et al., 2024b).

The dynamical analysis performed remains superficial, however, and further analysis of other heatwave mechanisms and longer time spans would be needed to conclude. Additionally, the ERA5 TXx5d for the 2021 PNW heatwave is of the same magnitude as E1; and while it is possible that the two distinct hypothesized distributions – the one giving events like E1 and E1, and the other giving events like the 2021 PNW heatwave and E2-13 – could overlap in terms of TXx5d, it questions the selection of E1 as distinct based on the jump in temperature anomaly between itself and the other parent events. The strong Pacific trough present next to the PNW region for E1 and E1 is also noticeable in certain other less intense parent events (see Appendix Figure A8) and even in the ERA5 heatwave (Figure 10), although to a substantially lesser degree, and with a blocking high placed slightly shifted from the exact PNW region studied in this paper.

The justification for removing E1 when estimating return period estimated for boosted analogues of the 2021 PNW heat-wave thus remains unclear. However, return period estimates calculated with and without boosted simulations from E1 seem consistent, with 95% confidence interval between  $10^3$  and  $10^4$  in both cases, and, importantly, remaining finite. This is not the case for estimates from the Large Ensembles only, where estimates range between 5000 years and  $\infty$ . The return period estimates using the boosting estimator also fit into the range of attributed return periods to the PNW 2021 shown in the review paper by White et al. (2023) which spans 200 years to  $\infty$ .

## 5 Conclusions



In this study, we develop a methodological framework, inspired by the iterative conditional probability chains described in Subset Sampling (Au and Beck, 2001; Finkel and O'Gorman, 2024), that can estimate unbiased, climatological probabilities of extreme events through Ensemble Boosting. We theoretically show that the boosting estimator is unbiased and that its relative errors are smaller than an equivalently expensive, brute force sampling estimator. The method is validated with a 1D red-noise process, for which we can give an expression for the window of time when more extreme events can be sampled. Using typical parameters of the CESM2 climate model, this expression gives remarkably precise estimates for both the lead time to select extremes and the intensity of the most extremes one can expect. The boosting estimator is evaluated on simulations from the fully-coupled climate model CESM2, where we show that it can accurately estimate the probability of very extreme events. We also show that the quality of these estimations depends on the number of parent events selected and the lead times used. Finally, as an application of the method, we estimate a return period for an event as intense as the record-shattering 2021 PNW heatwaye in the climate model.

The main findings can be listed as follows:

- 1. Return periods can be estimated for simulations generated through boosting, and their relative error is usually smaller than that of a naive estimator, for a sufficiently large return period. It is also approximately 85% cheaper to generate enough ensemble-boosted simulations to estimate return periods robustly than it is to do the same for climate model simulations using a naive estimator.
- 2. We provide a formula to estimate an order of magnitude of the lead time to use in practical cases where the dynamics of the extreme studied can be approximated by a red-noise process (Equation 12). In general, we show that return periods can be estimated more accurately and robustly when pooling boosted simulations for a range of longer lead times. In this study, this corresponds to −18 to −13 days before the event. These lead times nevertheless represent a somewhat empirical optimum between simulation independence and the likelihood of sampling more extreme events. Additionally, increasing the number of parent events in the Parent Ensemble is shown to also improve accuracy and robustness of return period estimates, by better estimating the theoretical probability P(T ≥ T<sub>ref</sub>) and sampling the set of antecedent conditions of the extreme studied. Since we find a weak relationship between the intensity of parent events and that of its boosted simulations, we recommend to sample a large diversity of parent events from the original climate model simulation.
- 3. Ensemble Boosting can be used as a tool to estimate return periods for real-life events like the 2021 PNW heatwave in a model context, conditional on the fact that the above parameter recommendations can be fulfilled. It can additionally be seen to be more efficient and robust compared to a naive estimator of a long climate simulation. However, caution needs to be taken when interpreting beyond the model world, and the numerical model's representation of extreme events and their frequency is required to be sufficiently similar to that of the real-world.

- 680 Code and data availability. The ERA5 re-analysis and the 100-member CESM2 data are publicly available:
  - CESM2: https://www.cesm.ucar.edu/community-projects/lens2/data-sets (Rodgers et al., 2021)
  - ERA5: https://doi.org/10.24381/cds.adbb2d47 (Hersbach et al., 2020)

Pre-processed data (CESM2 and ERA5) is available at https://doi.org/10.3929/ethz-b-000720049.

All code (preprocessing, calculation and plots) is available at https://github.com/luna-bloin/Boosting\_estimator.

## 685 Appendix A: Statistical properties of the boosting estimator

## A1 Unbiasedness of the boosting estimator


To estimate the expectation of the boosting estimator 10, we make the assumption that on the right hand side, the estimator  $\hat{p}_{T \geq T_{\text{ref}}}$  is independent from the ratio  $\frac{\hat{p}_{T \geq T_{\text{ext}}|AC_t^e}}{\hat{p}_{T \geq T_{\text{ref}}|AC_t^e}}$ . In other words, we assume that the probability to reach  $T_{\text{ref}}$  in the parent ensemble is independent from how more likely it is to reach  $T_{\text{ext}}$  than to reach  $T_{\text{ref}}$  in the boosted ensemble. We make a second approximation, which is that in the boosted ensemble the  $(T_b^m)_{1 \leq m \leq N_b}$  are independent one from another, see the main text for a discussion of these two hypotheses.

Under these hypotheses, the expectation of the boosting estimator is:

$$\mathbb{E}[\hat{p}_{T \ge T_{\text{ext}}}] = \mathbb{E}[\hat{p}_{T \ge T_{\text{ref}}}] \mathbb{E}\begin{bmatrix} \hat{p}_{T \ge T_{\text{ext}} | \text{AC}_t^{\epsilon}} \\ \hat{p}_{T \ge T_{\text{ref}} | \text{AC}_t^{\epsilon}} \end{bmatrix}. \tag{A1}$$

The first term on the right hand side is easily estimated:

$$\mathbb{E}[\hat{p}_{T \geq T_{\text{ref}}}] = p_{T \geq T_{\text{ref}}}.$$
 (A2)

Let us note  $\mathcal{N} := \sum_{m=0}^{N_b} \mathbb{1}(T_b^m \geq T_{\text{ext}})$  and  $\mathcal{D} := \sum_{m=0}^{N_b} \mathbb{1}(T_b^m \geq T_{\text{ref}})$  the numerator and denominator of the ratio on the right hand side. The expectation of the ratio can then be approximated, by a Taylor expansion, as (Kendall and others, 1948):

$$\mathbb{E}\left[\frac{\mathcal{N}}{\mathcal{D}}\right] = \frac{\mathbb{E}[\mathcal{N}]}{\mathbb{E}[\mathcal{D}]} \left( 1 - \frac{\operatorname{Cov}[\mathcal{N}, \mathcal{D}]}{\mathbb{E}[\mathcal{N}]\mathbb{E}[\mathcal{D}]} + \frac{\mathbb{V}[\mathcal{D}]}{\mathbb{E}[\mathcal{D}]^2} \right). \tag{A3}$$

The terms in this equation can be estimated independently:

$$\mathbb{E}[\mathcal{N}] = N_b \mathbb{E}[\hat{p}_{T \ge T_{\text{ext}}}] = N_b \mathbb{P}(T \ge T_{\text{ext}} \mid AC_t^{\epsilon}) = N_b p_{T > T_{\text{ext}} \mid AC_t^{\epsilon}}$$
(A4)

$$\mathbb{E}[\mathcal{D}] = N_b \mathbb{E}[\hat{p}_{T > T_{\text{ref}}}] = N_b \mathbb{P}(T \ge T_{\text{ref}} \mid AC_t^{\epsilon}) = N_b p_{T > T_{\text{ref}} \mid AC_t^{\epsilon}}$$
(A5)

$$\mathbb{V}[\mathcal{D}] = \sum_{m=0}^{N_b} \mathbb{V}[\mathbb{1}(T_b^m \ge T_{\text{ref}})] = N_b p_{T \ge T_{\text{ref}}|AC_t^{\epsilon}} (1 - p_{T \ge T_{\text{ref}}|AC_t^{\epsilon}})$$
(A6)

$$\mathbb{E}[\mathcal{ND}] = \sum_{m,\tilde{m}=0}^{N_b} \mathbb{E}[\mathbb{1}(T_b^m \ge T_{\text{ext}}) \mathbb{1}(T_b^{\tilde{m}} \ge T_{\text{ref}})]. \tag{A7}$$

For the last equation, one can separate the cases:

- there are  $N_b$  cases where n = m:

$$\mathbb{E}[\mathbb{1}(T_b^m \ge T_{\text{ext}})\mathbb{1}(T_b^m \ge T_{\text{ref}})] = \mathbb{E}[\mathbb{1}(T_b^m \ge T_{\text{ext}})] = p_{T > T_{\text{ext}}|\text{AC}_t^{\epsilon}}$$
(A8)

because  $T_{\text{ext}} \geq T_{\text{ref}}$ ,

- there are  $N_b(N_b-1)$  cases where  $m \neq \tilde{m}$ , using the independence assumption:

$$\mathbb{E}[\mathbb{1}(T_b^m \ge T_{\text{ext}})\mathbb{1}(T_b^{\tilde{m}} \ge T_{\text{ref}})] = \mathbb{E}[\mathbb{1}(T_b^m \ge T_{\text{ext}})]\mathbb{E}[\mathbb{1}(T_b^{\tilde{m}} \ge T_{\text{ref}})] = p_{T > T_{\text{ext}} \mid AC_t^{\epsilon}} p_{T > T_{\text{ref}} \mid AC_t^{\epsilon}}. \tag{A9}$$

In the end, this gives:

$$\mathbb{E}[\mathcal{N}\mathcal{D}] = N_b p_{T \ge T_{\text{ext}}|\text{AC}_t^{\epsilon}} (1 + (N_b - 1) p_{T \ge T_{\text{ref}}|\text{AC}_t^{\epsilon}}). \tag{A10}$$

Therefore:

$$\frac{\mathbb{V}[\mathcal{D}]}{\mathbb{E}[\mathcal{D}]^2} = \frac{1 - p_{T \ge T_{\text{ref}} \mid AC_t^{\epsilon}}}{N_b p_{T \ge T_{\text{ref}} \mid AC_t^{\epsilon}}}$$
(A11)

and

$$\frac{\operatorname{Cov}[\mathcal{N}, \mathcal{D}]}{\mathbb{E}[\mathcal{N}]\mathbb{E}[\mathcal{D}]} = \frac{\mathbb{E}[\mathcal{N}\mathcal{D}]}{\mathbb{E}[\mathcal{N}]\mathbb{E}[\mathcal{D}]} - 1 = \frac{1 + (N_b - 1)p_{T \ge T_{\text{ref}}|\operatorname{AC}^{\epsilon}_t}}{N_b p_{T > T_{\text{ref}}|\operatorname{AC}^{\epsilon}_t}} - 1 = \frac{1 - p_{T \ge T_{\text{ref}}|\operatorname{AC}^{\epsilon}_t}}{N_b p_{T > T_{\text{ref}}|\operatorname{AC}^{\epsilon}_t}}.$$
(A12)

As a result:

$$\mathbb{E}\left[\frac{\mathcal{N}}{\mathcal{D}}\right] = \frac{\mathbb{E}[\mathcal{N}]}{\mathbb{E}[\mathcal{D}]},\tag{A13}$$

which shows that the boosting estimator is unbiased:

$$\mathbb{E}[\hat{p}_{T \ge T_{\text{ext}}}] = p_{T \ge T_{\text{ext}}}.\tag{A14}$$

## 720 A2 Variance of the boosting estimator

For the boosting estimator to be useful to sample extremes, one needs to show that the relative error made when using this estimator is better than when using a naive estimator on the initial simulation. We now estimate the variance of the boosting estimator. With the same independence argument as previously:

$$\mathbb{E}[\hat{p}_{T \ge T_{\text{ext}}}^2] = \mathbb{E}[\hat{p}_{T \ge T_{\text{ref}}}^2] \mathbb{E}[\frac{\mathcal{N}^2}{\mathcal{D}^2}]. \tag{A15}$$

The  $\mathbb{E}[\hat{p}_{T\geq T_{\mathrm{ref}}}^2]$  term can be computed the same way as for the  $\mathbb{E}[\mathcal{ND}]$  term previously, which gives:

$$\mathbb{E}[\hat{p}_{T \ge T_{\text{ref}}}^2] = \frac{p_{T \ge T_{\text{ref}}} \cdot (1 + (N - 1) \cdot p_{T \ge T_{\text{ref}}})}{N}.$$
(A16)

The ratio can then be computed using the same formula as previously:

$$\mathbb{E}\left[\frac{\mathcal{N}^2}{\mathcal{D}^2}\right] = \frac{\mathbb{E}[\mathcal{N}^2]}{\mathbb{E}[\mathcal{D}^2]} \left(1 - \frac{\operatorname{Cov}[\mathcal{N}^2, \mathcal{D}^2]}{\mathbb{E}[\mathcal{N}^2]\mathbb{E}[\mathcal{D}^2]} + \frac{\mathbb{V}[\mathcal{D}^2]}{\mathbb{E}[\mathcal{D}^2]^2}\right). \tag{A17}$$

The terms  $\mathbb{E}[\mathcal{N}^2]$  and  $\mathbb{E}[\mathcal{D}^2]$  can be estimated as previously:


730 
$$\mathbb{E}[\mathcal{N}^2] = N_b \cdot p_{T > T_{\text{ext}} \mid \text{AC}_{\epsilon}^{\epsilon}} \cdot (1 + (N_b - 1) \cdot p_{T > T_{\text{ext}} \mid \text{AC}_{\epsilon}^{\epsilon}})$$
(A18)

$$\mathbb{E}[\mathcal{D}^2] = N_b \cdot p_{T \ge T_{\text{ref}}|AC_t^{\epsilon}} \cdot (1 + (N_b - 1) \cdot p_{T \ge T_{\text{ref}}|AC_t^{\epsilon}}). \tag{A19}$$

The term  $\mathbb{E}[\mathcal{N}^2\mathcal{D}^2]$  is less straightforward. By the definition of the product:

$$\mathbb{E}[\mathcal{N}^2 \mathcal{D}^2] = \sum_{\tilde{m}, m, o, n=1}^{N_b} \mathbb{E}[\mathbb{1}(T_b^{\tilde{m}} \ge T_{\text{ext}}) \mathbb{1}(T_b^m \ge T_{\text{ext}}) \mathbb{1}(T_b^o \ge T_{\text{ref}}) \mathbb{1}(T_b^p \ge T_{\text{ref}})]. \tag{A20}$$

This sum has  $N^4$  terms that can be decomposed into four cases for the quadruplet  $(\tilde{m}, m, o, p)$ :

- two terms in the quadruplet are equal and the two others are different and different from one another, e.g.  $\tilde{m}=m=2$ , o=3 and p=4. Among them:
  - there are  $N_b(N_b-1)(N_b-2)$  such quadruplets for which  $\tilde{m}=m$  and the associated expectation for each of them is:

$$\mathbb{E}[\mathbb{1}(T_b^{\tilde{m}} \ge T_{\text{ext}})]\mathbb{E}[\mathbb{1}(T_b^o \ge T_{\text{ref}})]\mathbb{E}[\mathbb{1}(T_b^p \ge T_{\text{ref}})] = p_{T \ge T_{\text{ext}}|\text{AC}_t^{\epsilon}}(p_{T \ge T_{\text{ref}}|\text{AC}_t^{\epsilon}})^2$$
(A21)

- there are  $5N_b(N_b-1)(N_b-2)$  such quadruplets for which  $\tilde{m} \neq m$  and the associated expectation for each of them is:

$$\mathbb{E}[\mathbb{1}(T_b^{\tilde{m}} \ge T_{\text{ext}})] \mathbb{E}[\mathbb{1}(T_b^m \ge T_{\text{ext}})] \mathbb{E}[\mathbb{1}(T_b^o \ge T_{\text{ref}})] = (p_{T \ge T_{\text{ext}}|\text{AC}_t^{\epsilon}})^2 p_{T \ge T_{\text{ref}}|\text{AC}_t^{\epsilon}}$$
(A22)

- three terms in the quadruplet are equal and the last one is different from them, e.g.  $\tilde{m}=m=o\neq p$ . Among them:
  - there are  $2N_b(N_b-1)$  such quadruplets for which  $\tilde{m}=m$  and the associated expectation for each of them is:

$$\mathbb{E}[\mathbb{1}(T_h^{\tilde{m}} \ge T_{\text{ext}})]\mathbb{E}[\mathbb{1}(T_h^o \ge T_{\text{ref}})] = p_{T \ge T_{\text{ref}}|AC^{\epsilon}} p_{T \ge T_{\text{ref}}|AC^{\epsilon}}$$
(A23)

- there are  $2N_b(N_b-1)$  such quadruplets for which  $\tilde{m}\neq m$  and the associated expectation for each of them is:

$$\mathbb{E}[\mathbb{1}(T_b^{\tilde{m}} \ge T_{\text{ext}})]\mathbb{E}[\mathbb{1}(T_b^m \ge T_{\text{ext}})] = (p_{T \ge T_{\text{ext}}|\text{AC}_{\epsilon}^{\epsilon}})^2 \tag{A24}$$

- the four terms in the quadruplet are equal: there are  $N_b$  such quadruplets and the associated expectation is:

$$\mathbb{E}[\mathbb{1}(T_b^{\tilde{m}} \ge T_{\text{ext}})] = p_{T > T_{\text{ext}} \mid \text{AC}_t^{\epsilon}} \tag{A25}$$

- the four terms in the quadruplet are different: there are  $N_b^4 - 6N_b(N_b - 1)(N_b - 2) - 4N_b(N_b - 1) - N_b$  such quadruplets and the associated expectation is:

$$\mathbb{E}[\mathbb{1}(T_b^{\tilde{m}} \ge T_{\text{ext}})]\mathbb{E}[\mathbb{1}(T_b^m \ge T_{\text{ext}})]\mathbb{E}[\mathbb{1}(T_b^o \ge T_{\text{ref}})]\mathbb{E}[\mathbb{1}(T_b^p \ge T_{\text{ref}})] = (p_{T > T_{\text{ext}} \mid AC_t^{\epsilon}} p_{T > T_{\text{ref}} \mid AC_t^{\epsilon}})^2$$
(A26)

Thus, in the end:


$$\mathbb{E}[\mathcal{N}^{2}\mathcal{D}^{2}] = (N_{b}^{4} - 6N_{b}(N_{b} - 1)(N_{b} - 2) - 4N_{b}(N_{b} - 1) - N_{b}) (p_{T \geq T_{\text{ext}}|\text{AC}_{t}^{\epsilon}} p_{T \geq T_{\text{ref}}|\text{AC}_{t}^{\epsilon}})^{2} 
+ N_{b}(N_{b} - 1)(N_{b} - 2)p_{T \geq T_{\text{ext}}|\text{AC}_{t}^{\epsilon}} (p_{T \geq T_{\text{ref}}|\text{AC}_{t}^{\epsilon}})^{2} 
+ 5N_{b}(N_{b} - 1)(N_{b} - 2)(p_{T \geq T_{\text{ext}}|\text{AC}_{t}^{\epsilon}})^{2} p_{T \geq T_{\text{ref}}|\text{AC}_{t}^{\epsilon}} 
+ 2N_{b}(N_{b} - 1)p_{T \geq T_{\text{ext}}|\text{AC}_{t}^{\epsilon}} p_{T \geq T_{\text{ref}}|\text{AC}_{t}^{\epsilon}} 
+ 2N_{b}(N_{b} - 1)(p_{T \geq T_{\text{ext}}|\text{AC}_{t}^{\epsilon}})^{2} 
+ N_{b}p_{T \geq T_{\text{ext}}|\text{AC}_{t}^{\epsilon}} \tag{A27}$$

With a similar manner, one retrieves the fourth moment of  $\mathcal{D}$  (to compute the variance of  $\mathcal{D}^2$ ):

$$\mathbb{E}[\mathcal{D}^{4}] = (N_{b}^{4} - 6N_{b}(N_{b} - 1)(N_{b} - 2) - 2N_{b}(N_{b} - 1) - N_{b}) (p_{T \geq T_{\text{ref}}|AC_{t}^{\epsilon}})^{4}$$

$$+ 6N_{b}(N_{b} - 1)(N_{b} - 2)(p_{T \geq T_{\text{ref}}|AC_{t}^{\epsilon}})^{3}$$

$$+ 4N_{b}(N_{b} - 1)(p_{T \geq T_{\text{ref}}|AC_{t}^{\epsilon}})^{2}$$

$$+ N_{b}p_{T \geq T_{\text{ref}}|AC_{t}^{\epsilon}}.$$
(A28)

Using the formulas above, one can then give an expression for the variance of the boosting estimator and the relative error (of which we do not give a closed form here):

$$\mathbb{RE} := \frac{\sqrt{\mathbb{V}[\hat{p}_{T \ge T_{\text{ext}}}]}}{\mathbb{E}[\hat{p}_{T \ge T_{\text{ext}}}]} = \sqrt{\frac{\mathbb{E}[\hat{p}_{T \ge T_{\text{ext}}}]}{\mathbb{E}[\hat{p}_{T \ge T_{\text{ext}}}]^2} - 1}.$$
(A29)

## Appendix B: Ensemble boosting with the Orstein-Uhlenbeck process

Starting an ensemble of boosted members from a parent reaching the value  $x_0$  with the Orstein-Uhlenbeck process, it can be shown (Risken, 1996) that the distribution of this ensemble is Gaussian with a mean m evolving as:

$$m(t) = x_0 e^{-\alpha t}$$
 (B1)

and a variance  $s^2$ :

$$s^{2}(t) = \frac{\sigma^{2}}{2\alpha}(1 - e^{-2\alpha t}). \tag{B2}$$

An example of the evolution of such an ensemble can be seen in Figure 4a. At the limit  $t \to \infty$ , the mean decreases back to 0, while the variance reaches the stationary variance  $\sigma^2/2\alpha$ . This does not allow to sample extremes with a limited sample size. However, in the short term the variance increases faster than the mean decreases, which opens a window where the boosted ensemble can reach higher values than its starting point  $x_0$ . We now show this mechanism more formally. We consider the evolution of the mean plus a number k > 0 of standard deviations, where k would typically be a function of the number of members in the boosted ensemble:

$$f(t) := m(t) + ks(t) = x_0 e^{-\alpha t} + \frac{k\sigma}{\sqrt{2\alpha}} \sqrt{1 - e^{-2\alpha t}}.$$
 (B3)

We find the time  $t^*$  where f reaches its maximum by setting its derivative to 0:

$$f'(t^*) = -\alpha x_0 e^{-\alpha t^*} + \frac{k\sigma}{\sqrt{2\alpha}} \frac{\alpha e^{-2\alpha t^*}}{\sqrt{1 - e^{-2\alpha t^*}}} = 0,$$
(B4)

which admits as solution:

$$t^* = \frac{1}{2\alpha} \ln(1 + k^2 \frac{\sigma^2 / 2\alpha}{x_0^2}). \tag{B5}$$

The maximum value reached by f is then:

$$f(t^*) = \sigma^2 / 2\alpha \sqrt{(\frac{x_0}{\sigma^2 / 2\alpha})^2 + k^2}$$
 (B6)

Author contributions. LBW: conception, formal analysis, methodology, data curation, investigation, software, writing (original draft preparation, review & editing); RN: conception, formal analysis, methodology, supervision, writing (review & editing); VH: conception (original idea), methodology, investigation, writing (review & editing); UB: resources, data curation, software (climate model simulations), writing (review & editing); EF: funding acquisition, writing (review & editing); RK: funding acquisition, writing (review & editing)

Competing interests. The authors declare that one of the co-authors is a member of the editorial board of WCD (Erich Fischer).

Acknowledgements. We gratefully acknowledge funding from the EU Horizon 2020 Project XAIDA (grant agreement 101003469). LBW and RK are part of the SPEED2ZERO, a Joint Initiative co-financed by the ETH Board. The authors would like to thank Lukas Papritz and Belinda Hotz, for providing the detrended ERA5 temperature time series. We also thank all contributors, including the National Center for

Figure A1. Density distribution of boosted yearly summer maximum of daily maximum temperature anomalies with a running mean of 5 days (TXx5d) compared to that of the climatology. Climatological distribution (in black) is derived from the control period, while the distribution of boosted simulations at lead time -12 (in orange) comes from one parent event. The magnitude of the parent event is highlighted (in blue).

Atmospheric Research (NCAR) for developing the Community Earth System Model, and the IBS Center for Climate Physics in South Korea for running the 100-member Large Ensemble CESM2 data set. Furthermore, since all analysis was carried out in Python, we thank all its contributors, as well as those who contributed to Python packages, in particular xarray, numpy and scipy. Finally, we would like to thank the anonymous referees for their useful comments, in particular C. Martínez-Villalobos for suggesting to validate the theoretical methodology with a red-noise process.

Figure A2. Return periods of TXx5d in the PNW region estimated with the boosting estimator under stationary climate conditions. Estimated return periods with the boosting algorithm for parent events selected from test slice 1 of the pre-industrial control simulation, with perturbation lead times from -18 (a) to -7 (l) days, are shown in orange. In each panel, the TXx5d of the 4000 years of the pre-industrial control simulation are shown in black. The TXx5d of the 50-year test slice is shown in blue. The five selected parent events are highlighted in diamonds. The vertical dashed orange line represents the reference temperature  $T_{\rm ref}$  in each test slice. For the return periods of the control and test slice simulations, a GEV law is fitted and the estimated return period is shown (solid line) with a bootstrap 95% confidence interval (shaded). For the boosted simulations, the shaded area shows the bootstrap 95% confidence interval (see Methods).

Figure A3. As Figure A2 for test slice 2.

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

Figure A4. Return periods of TXx5d in the PNW region estimated with the boosting estimator under stationary climate conditions. Estimated return periods with the boosting algorithm for parent events selected from test slice 1 + 2 of the pre-industrial control simulation are shown in orange. Perturbation lead times are pooled together from (a) -18 to -13 days and (b) -12 to -7 days. In each panel, the TXx5d of the 4000 years of the pre-industrial control simulation are shown in black. The TXx5d of the 100-year test slice is shown in blue. The 10 selected parent events are highlighted in diamonds. The horizontal dashed orange line represents the reference temperature  $T_{\text{ref}}$  in each test slice. For the return periods of the control and test slice simulations, a GEV law is fitted and the estimated return period is shown (solid line) with a bootstrap 95% confidence interval (shaded). For the boosted simulations, the shaded area shows the bootstrap 95% confidence interval (see Methods).

Bartusek, S., Kornhuber, K., and Ting, M.: 2021 North American heatwave amplified by climate change-driven nonlinear interactions, Nature Climate Change, 12, 1143–1150, https://doi.org/10.1038/s41558-022-01520-4, publisher: Nature Publishing Group, 2022.

Coles, S., Bawa, J., Trenner, L., and Dorazio, P.: An introduction to statistical modeling of extreme values, vol. 208, Springer, 2001.



Cooley, D.: Return Periods and Return Levels Under Climate Change, in: Extremes in a Changing Climate: Detection, Analysis and Uncertainty, edited by AghaKouchak, A., Easterling, D., Hsu, K., Schubert, S., and Sorooshian, S., pp. 97–114, Springer Netherlands, Dordrecht, ISBN 978-94-007-4479-0, https://doi.org/10.1007/978-94-007-4479-0\_4, 2013.

Danabasoglu, G., Lamarque, J., Bacmeister, J., Bailey, D. A., DuVivier, A. K., Edwards, J., Emmons, L. K., Fasullo, J., Garcia, R., Gettelman, A., Hannay, C., Holland, M. M., Large, W. G., Lauritzen, P. H., Lawrence, D. M., Lenaerts, J. T. M., Lindsay, K., Lipscomb, W. H., Mills, M. J., Neale, R., Oleson, K. W., Otto-Bliesner, B., Phillips, A. S., Sacks, W., Tilmes, S., Van Kampenhout, L., Vertenstein, M., Bertini, A., Dennis, J., Deser, C., Fischer, C., Fox-Kemper, B., Kay, J. E., Kinnison, D., Kushner, P. J., Larson, V. E., Long, M. C., Mickelson, S., Moore, J. K., Nienhouse, E., Polvani, L., Rasch, P. J., and Strand, W. G.: The Community Earth System Model Version 2 (CESM2), Journal of Advances in Modeling Earth Systems, 12, e2019MS001916, https://doi.org/10.1029/2019MS001916, 2020.

Figure A5. Spearman correlation between TXx5d of parent event and the 90th percentile of TXx5d among its boosted simulations.

- Dunne, J. P., Stouffer, R. J., and John, J. G.: Reductions in labour capacity from heat stress under climate warming, Nature Climate Change, 3, 563–566, https://doi.org/10.1038/nclimate1827, publisher: Nature Publishing Group, 2013.
- Finkel, J. and O'Gorman, P. A.: Bringing Statistics to Storylines: Rare Event Sampling for Sudden, Transient Extreme Events, Journal of Advances in Modeling Earth Systems, 16, e2024MS004 264, https://doi.org/10.1029/2024MS004264, 2024.
  - Fischer, E. M., Sippel, S., and Knutti, R.: Increasing probability of record-shattering climate extremes, Nature Climate Change, 11, 689–695, https://doi.org/10.1038/s41558-021-01092-9, 2021.
  - Fischer, E. M., Beyerle, U., Bloin-Wibe, L., Gessner, C., Humphrey, V., Lehner, F., Pendergrass, A. G., Sippel, S., Zeder, J., and Knutti, R.: Storylines for unprecedented heatwaves based on ensemble boosting, Nature Communications, 14, 4643, https://doi.org/10.1038/s41467-023-40112-4, 2023.


- Galfi, V. M. and Lucarini, V.: Fingerprinting Heatwaves and Cold Spells and Assessing Their Response to Climate Change Using Large Deviation Theory, Physical Review Letters, 127, 058701, https://doi.org/10.1103/PhysRevLett.127.058701, 2021.
- Gessner, C., Fischer, E. M., Beyerle, U., and Knutti, R.: Very rare heat extremes: quantifying and understanding using ensemble reinitialization, Journal of Climate, pp. 1–46, https://doi.org/10.1175/JCLI-D-20-0916.1, 2021.
- Gessner, C., Fischer, E. M., Beyerle, U., and Knutti, R.: Multi-year drought storylines for Europe and North America from an iteratively perturbed global climate model, Weather and Climate Extremes, 38, 100 512, https://doi.org/10.1016/j.wace.2022.100512, 2022.
  - Gessner, C., Fischer, E. M., Beyerle, U., and Knutti, R.: Developing Low-Likelihood Climate Storylines for Extreme Precipitation Over Central Europe, Earth's Future, 11, e2023EF003628, https://doi.org/10.1029/2023EF003628, 2023.

Figure A6. Return periods of TXx5d in the PNW region estimated with the boosting estimator for non-stationary (climate change) conditions. Estimated return periods with the boosting algorithm for all parent events in the Parent Ensemble, with perturbation lead times from -18 (a) to -7 (I) days, are shown in orange. The TXx5d of the (black dots) 100-member and (blue dots) 30-member Large Ensemble, with a fitted GEV law and shown with the median (solid line) and a bootstrapped 95% confidence interval (shaded area). The parent events in the Parent Ensemble are shown with blue diamonds, and the most extreme parent E1, and the most extreme event in the 100-member Large Ensemble  $\tilde{E}1$  are highlighted with stars. The orange horizontal line denotes the reference TXx5d  $T_{\rm ref}$ , while the black horizontal line denotes the 2021 ERA5 TXx5d.

Giardina, C., Kurchan, J., Lecomte, V., and Tailleur, J.: Simulating Rare Events in Dynamical Processes, Journal of Statistical Physics, 145, 787–811, https://doi.org/10.1007/s10955-011-0350-4, 2011.


Figure A7. Return periods of TXx5d in the PNW region estimated with the boosting estimator for non-stationary (climate change) conditions for a lead time range of -18 to -13 days. Estimated return periods with the boosting algorithm for (a) all parent events in the Parent Ensemble, and (b-h) E1 - E13 are shown in orange. The TXx5d of the (black dots) 100-member and (blue dots) 30-member Large Ensemble, with a fitted GEV law and shown with the median (solid line) and a bootstrapped 95% confidence interval (shaded area). The parent events in the Parent Ensemble are shown with blue diamonds, and the most extreme parent E1, and the most extreme event in the 100-member Large Ensemble  $\tilde{E}1$  are highlighted with stars. The orange horizontal line denotes the reference TXx5d  $T_{\rm ref}$ , while the black horizontal line denotes the 2021 ERA5 TXx5d.

Figure A8. Tx5d anomaly [ $^{\circ}$ C] and Z500 [m] contour lines on the peak day of each heatwave, for the top 13 most extreme events (a-m) E1 to E13 in the 30-member Large Ensemble. The black box indicates the region of interest.

Giardinà, C., Kurchan, J., and Peliti, L.: Direct Evaluation of Large-Deviation Functions, Physical Review Letters, 96, 120 603, https://doi.org/10.1103/PhysRevLett.96.120603, publisher: American Physical Society, 2006.

Gourdji, S. M., Sibley, A. M., and Lobell, D. B.: Global crop exposure to critical high temperatures in the reproductive period: historical trends and future projections, Environmental Research Letters, 8, 024 041, https://doi.org/10.1088/1748-9326/8/2/024041, publisher: IOP Publishing, 2013.


Hersbach, H., Bell, B., Berrisford, P., Hirahara, S., Horányi, A., Muñoz-Sabater, J., Nicolas, J., Peubey, C., Radu, R., Schepers, D., Simmons, A., Soci, C., Abdalla, S., Abellan, X., Balsamo, G., Bechtold, P., Biavati, G., Bidlot, J., Bonavita, M., De Chiara, G., Dahlgren, P., Dee, D., Diamantakis, M., Dragani, R., Flemming, J., Forbes, R., Fuentes, M., Geer, A., Haimberger, L., Healy, S., Hogan, R. J.,

Figure A9. As Figure 10 for the 100-member Large Ensemble.



Hólm, E., Janisková, M., Keeley, S., Laloyaux, P., Lopez, P., Lupu, C., Radnoti, G., de Rosnay, P., Rozum, I., Vamborg, F., Villaume, S., and Thépaut, J.-N.: The ERA5 global reanalysis, Quarterly Journal of the Royal Meteorological Society, 146, 1999–2049, https://doi.org/10.1002/qj.3803, \_eprint: https://onlinelibrary.wiley.com/doi/pdf/10.1002/qj.3803, 2020.

Horton, R. M., Mankin, J. S., Lesk, C., Coffel, E., and Raymond, C.: A Review of Recent Advances in Research on Extreme Heat Events, Current Climate Change Reports, 2, 242–259, https://doi.org/10.1007/s40641-016-0042-x, 2016.

Kendall, M. G. and others: The advanced theory of statistics. Vols. 1., The advanced theory of statistics. Vols. 1., 1, publisher: Charles Griffin and Co., Ltd., 42 Drury Lane, London, 1948.

Krishnamurthy, V.: Predictability of Weather and Climate, Earth and Space Science, 6, 1043–1056, https://doi.org/10.1029/2019EA000586, eprint: https://onlinelibrary.wiley.com/doi/pdf/10.1029/2019EA000586, 2019.

Lorenz, E. N.: Predictability – a problem partly solved, in: Predictability of Weather and Climate, edited by Palmer, T. and Hagedorn, R., pp. 40–58, Cambridge University Press, 1 edn., ISBN 978-0-521-84882-4 978-0-511-61765-2 978-1-107-41485-3, https://doi.org/10.1017/CBO9780511617652.004, 2006.

Lucarini, V., Galfi, V. M., Riboldi, J., and Messori, G.: Typicality of the 2021 Western North America summer heatwave, Environmental Research Letters, 18, 015 004, https://doi.org/10.1088/1748-9326/acab77, publisher: IOP Publishing, 2023.

Lüthi, S., Huber, V., Pascal, M., Beyerle, U., Pyrina, M., Domeisen, D., Vicedo-Cabrera, A. M., and Fischer, E.: Storylines for month-long heatwaves and associated heat-related mortality impacts over Western Europe, https://doi.org/10.21203/rs.3.rs-5356341/v1, 2024.

Malinina, E. and Gillett, N. P.: The 2021 heatwave was less rare in Western Canada than previously thought, Weather and Climate Extremes, 43, 100 642, https://doi.org/10.1016/j.wace.2024.100642, 2024.

Figure A10. As Figure A8 for the 100-member Large Ensemble.

McKinnon, K. A. and Simpson, I. R.: How Unexpected Was the 2021 Pacific Northwest Heatwave?, Geophysical Research Letters, 49, e2022GL100380, https://doi.org/10.1029/2022GL100380, \_eprint: https://onlinelibrary.wiley.com/doi/pdf/10.1029/2022GL100380, 2022.

Meehl, G. A. and Tebaldi, C.: More Intense, More Frequent, and Longer Lasting Heat Waves in the 21st Century, Science, 305, 994–997, https://doi.org/10.1126/science.1098704, publisher: American Association for the Advancement of Science, 2004.

Miloshevich, G., Cozian, B., Abry, P., Borgnat, P., and Bouchet, F.: Probabilistic forecasts of extreme heatwaves using convolutional neural networks in a regime of lack of data, Physical Review Fluids, 8, 040 501, https://doi.org/10.1103/PhysRevFluids.8.040501, publisher: American Physical Society, 2023.

- Miralles, D. G., Teuling, A. J., Van Heerwaarden, C. C., and Vilà-Guerau De Arellano, J.: Mega-heatwave temperatures due to combined soil desiccation and atmospheric heat accumulation, Nature Geoscience, 7, 345–349, https://doi.org/10.1038/ngeo2141, 2014.
  - Neal, E., Huang, C. S. Y., and Nakamura, N.: The 2021 Pacific Northwest Heat Wave and Associated Blocking: Meteorology and the Role of an Upstream Cyclone as a Diabatic Source of Wave Activity, Geophysical Research Letters, 49, e2021GL097699, https://doi.org/10.1029/2021GL097699, eprint: https://onlinelibrary.wiley.com/doi/pdf/10.1029/2021GL097699, 2022.
- 870 Noyelle, R.: Statistical and dynamical aspects of extreme heatwaves in the mid-latitudes, 2024.

895

- Noyelle, R., Robin, Y., Naveau, P., Yiou, P., and Faranda, D.: Integration of physical bound constraints to alleviate shortcomings of statistical models for extreme temperatures, https://hal.science/hal-04479249, 2024a.
- Noyelle, R., Yiou, P., and Faranda, D.: Investigating the typicality of the dynamics leading to extreme temperatures in the IPSL-CM6A-LR model, Climate Dynamics, 62, 1329–1357, https://doi.org/10.1007/s00382-023-06967-5, 2024b.
- 875 Otto, F. E. L., Massey, N., van Oldenborgh, G. J., Jones, R. G., and Allen, M. R.: Reconciling two approaches to attribution of the 2010 Russian heat wave, Geophysical Research Letters, 39, https://doi.org/10.1029/2011GL050422, \_eprint: https://onlinelibrary.wiley.com/doi/pdf/10.1029/2011GL050422, 2012.
  - Overland, J. E.: Causes of the Record-Breaking Pacific Northwest Heatwave, Late June 2021, Atmosphere, 12, 1434, https://doi.org/10.3390/atmos12111434, number: 11 Publisher: Multidisciplinary Digital Publishing Institute, 2021.
- Perkins, S. E.: A review on the scientific understanding of heatwaves—Their measurement, driving mechanisms, and changes at the global scale, Atmospheric Research, 164-165, 242–267, https://doi.org/10.1016/j.atmosres.2015.05.014, 2015.
  - Pfahl, S. and Wernli, H.: Quantifying the relevance of atmospheric blocking for co-located temperature extremes in the Northern Hemisphere on (sub-)daily time scales: BLOCKING AND TEMPERATURE EXTREMES, Geophysical Research Letters, 39, n/a–n/a, https://doi.org/10.1029/2012GL052261, 2012.
- Philip, S., Kew, S., van Oldenborgh, G. J., Otto, F., Vautard, R., van der Wiel, K., King, A., Lott, F., Arrighi, J., Singh, R., and van Aalst, M.: A protocol for probabilistic extreme event attribution analyses, Advances in Statistical Climatology, Meteorology and Oceanography, 6, 177–203, https://doi.org/10.5194/ascmo-6-177-2020, publisher: Copernicus GmbH, 2020.
  - Philip, S. Y., Kew, S. F., van Oldenborgh, G. J., Anslow, F. S., Seneviratne, S. I., Vautard, R., Coumou, D., Ebi, K. L., Arrighi, J., Singh, R., van Aalst, M., Pereira Marghidan, C., Wehner, M., Yang, W., Li, S., Schumacher, D. L., Hauser, M., Bonnet, R., Luu, L. N., Lehner, F., Gillett, N., Tradovsky, J. S., Vacchi, G. A., Podell, C., Stull, P. R., Howard, P., and Otto, F. E. L., Papid attribution analysis of the extraordinary
- N., Tradowsky, J. S., Vecchi, G. A., Rodell, C., Stull, R. B., Howard, R., and Otto, F. E. L.: Rapid attribution analysis of the extraordinary heat wave on the Pacific coast of the US and Canada in June 2021, Earth System Dynamics, 13, 1689–1713, https://doi.org/10.5194/esd-13-1689-2022, publisher: Copernicus GmbH, 2022.
  - Plotkin, D. A., Webber, R. J., O'Neill, M. E., Weare, J., and Abbot, D. S.: Maximizing Simulated Tropical Cyclone Intensity With Action Minimization, Journal of Advances in Modeling Earth Systems, 11, 863–891, https://doi.org/10.1029/2018MS001419, \_eprint: https://onlinelibrary.wiley.com/doi/pdf/10.1029/2018MS001419, 2019.
  - Ragone, F., Wouters, J., and Bouchet, F.: Computation of extreme heat waves in climate models using a large deviation algorithm, Proceedings of the National Academy of Sciences, 115, 24–29, https://doi.org/10.1073/pnas.1712645115, 2018.
  - Rahmstorf, S. and Coumou, D.: Increase of extreme events in a warming world, Proceedings of the National Academy of Sciences, 108, 17 905–17 909, https://doi.org/10.1073/pnas.1101766108, 2011.
- 900 Ranasinghe, R., Ruane, A., Vautard, R., Arnell, N., Coppola, E., Cruz, F., Dessai, S., Islam, A., Rahimi, M., Ruiz Carrascal, D., Sillmann, J., Sylla, M., Tebaldi, C., Wang, W., and Zaaboul, R.: Climate Change Information for Regional Impact and for Risk Assessment, in: Climate Change 2021: The Physical Science Basis. Contribution of Working Group I to the Sixth Assessment Report of the

- Intergovernmental Panel on Climate Change, edited by Masson-Delmotte, V., Zhai, P., Pirani, A., Connors, S. L., Péan, C., Berger, S., Caud, N., Chen, Y., Goldfarb, L., Gomis, M. I., Huang, M., Leitzell, K., Lonnoy, E., Matthews, J. B. R., Maycock, T. K., Waterfield, T., Yelekçi, O., Yu, R., and Zhou, B., pp. 1767–1925, Cambridge University Press, Cambridge, UK and New York, NY, USA, https://doi.org/10.1017/9781009157896.014, section: 12 Type: Book Section, 2021.
  - Risken, H.: The Fokker-Planck Equation: Methods of Solution and Applications, vol. 18 of *Springer Series in Synergetics*, Springer Berlin Heidelberg, Berlin, Heidelberg, ISBN 978-3-540-61530-9 978-3-642-61544-3, https://doi.org/10.1007/978-3-642-61544-3, 1996.
- Robine, J.-M., Cheung, S. L. K., Le Roy, S., Van Oyen, H., Griffiths, C., Michel, J.-P., and Herrmann, F. R.: Death toll exceeded 70,000 in Europe during the summer of 2003, Comptes Rendus Biologies, 331, 171–178, https://doi.org/10.1016/j.crvi.2007.12.001, 2008.
  - Rodgers, K. B., Lee, S.-S., Rosenbloom, N., Timmermann, A., Danabasoglu, G., Deser, C., Edwards, J., Kim, J.-E., Simpson, I. R., Stein, K., Stuecker, M. F., Yamaguchi, R., Bódai, T., Chung, E.-S., Huang, L., Kim, W. M., Lamarque, J.-F., Lombardozzi, D. L., Wieder, W. R., and Yeager, S. G.: Ubiquity of human-induced changes in climate variability, Earth System Dynamics, 12, 1393–1411, https://doi.org/10.5194/esd-12-1393-2021, publisher: Copernicus GmbH, 2021.
- Patrice Röthlisberger, M. and Papritz, L.: Quantifying the physical processes leading to atmospheric hot extremes at a global scale, Nature Geoscience, 16, 210–216, https://doi.org/10.1038/s41561-023-01126-1, publisher: Nature Publishing Group, 2023.
  - Schaller, N., Sillmann, J., Anstey, J., Fischer, E. M., Grams, C. M., and Russo, S.: Influence of blocking on Northern European and Western Russian heatwaves in large climate model ensembles, Environmental Research Letters, 13, 054015, https://doi.org/10.1088/1748-9326/aaba55, publisher: IOP Publishing, 2018.
- 920 Schumacher, D. L., Hauser, M., and Seneviratne, S. I.: Drivers and Mechanisms of the 2021 Pacific Northwest Heatwave, Earth's Future, 10, e2022EF002 967, https://doi.org/10.1029/2022EF002967, \_eprint: https://onlinelibrary.wiley.com/doi/pdf/10.1029/2022EF002967, 2022.
  - Seneviratne, S., Zhang, X., Adnan, M., Badi, W., Dereczynski, C., Di Luca, A., Ghosh, S., Iskandar, I., Kossin, J., Lewis, S., Otto, F., Pinto, I., Satoh, M., Vicente-Serrano, S., Wehner, M., and Zhou, B.: Weather and Climate Extreme Events in a Changing Climate, in: Climate Change 2021: The Physical Science Basis. Contribution of Working Group I to the Sixth Assessment Report of the In-
- tergovernmental Panel on Climate Change, edited by Masson-Delmotte, V., Zhai, P., Pirani, A., Connors, S. L., Péan, C., Berger, S., Caud, N., Chen, Y., Goldfarb, L., Gomis, M. I., Huang, M., Leitzell, K., Lonnoy, E., Matthews, J. B. R., Maycock, T. K., Waterfield, T., Yelekçi, O., Yu, R., and Zhou, B., pp. 1513–1765, Cambridge University Press, Cambridge, UK and New York, NY, USA, https://doi.org/10.1017/9781009157896.013, section: 11 Type: Book Section, 2021.
- Shepherd, T. G., Boyd, E., Calel, R. A., Chapman, S. C., Dessai, S., Dima-West, I. M., Fowler, H. J., James, R., Maraun, D., Martius, O., Senior, C. A., Sobel, A. H., Stainforth, D. A., Tett, S. F. B., Trenberth, K. E., Van Den Hurk, B. J. J. M., Watkins, N. W., Wilby, R. L., and Zenghelis, D. A.: Storylines: an alternative approach to representing uncertainty in physical aspects of climate change, Climatic Change, 151, 555–571, https://doi.org/10.1007/s10584-018-2317-9, 2018.
  - Suarez-Gutierrez, L., Li, C., Müller, W. A., and Marotzke, J.: Internal variability in European summer temperatures at 1.5 °C and 2 °C of global warming, Environmental Research Letters, 13, 064 026, https://doi.org/10.1088/1748-9326/aaba58, publisher: IOP Publishing, 2018.

935

940

Thiery, W., Lange, S., Rogelj, J., Schleussner, C.-F., Gudmundsson, L., Seneviratne, S. I., Andrijevic, M., Frieler, K., Emanuel, K., Geiger, T., Bresch, D. N., Zhao, F., Willner, S. N., Büchner, M., Volkholz, J., Bauer, N., Chang, J., Ciais, P., Dury, M., François, L., Grillakis, M., Gosling, S. N., Hanasaki, N., Hickler, T., Huber, V., Ito, A., Jägermeyr, J., Khabarov, N., Koutroulis, A., Liu, W., Lutz, W., Mengel, M., Müller, C., Ostberg, S., Reyer, C. P. O., Stacke, T., and Wada, Y.: Intergenerational inequities in exposure to climate extremes, Science, 374, 158–160, https://doi.org/10.1126/science.abi7339, publisher: American Association for the Advancement of Science, 2021.

- Trevisan, A. and Palatella, L.: Chaos and weather forecasting: the role of the unstable subspace in predictability and state estimation problems, International Journal of Bifurcation and Chaos, 21, 3389–3415, https://doi.org/10.1142/S0218127411030635, publisher: World Scientific Publishing Co., 2011.
- Vannitsem, S.: Predictability of large-scale atmospheric motions: Lyapunov exponents and error dynamics, Chaos: An Interdisciplinary Journal of Nonlinear Science, 27, 032 101, https://doi.org/10.1063/1.4979042, 2017.
  - Vicedo-Cabrera, A. M., Scovronick, N., Sera, F., Royé, D., Schneider, R., Tobias, A., Astrom, C., Guo, Y., Honda, Y., Hondula, D. M., Abrutzky, R., Tong, S., Coelho, M. d. S. Z. S., Saldiva, P. H. N., Lavigne, E., Correa, P. M., Ortega, N. V., Kan, H., Osorio, S., Kyselý, J., Urban, A., Orru, H., Indermitte, E., Jaakkola, J. J. K., Ryti, N., Pascal, M., Schneider, A., Katsouyanni, K., Samoli, E., Mayvaneh, F., Entezari, A., Goodman, P., Zeka, A., Michelozzi, P., de'Donato, F., Hashizume, M., Alahmad, B., Diaz, M. H., Valencia, C. D. L. C.,
- Overcenco, A., Houthuijs, D., Ameling, C., Rao, S., Di Ruscio, F., Carrasco-Escobar, G., Seposo, X., Silva, S., Madureira, J., Holobaca, I. H., Fratianni, S., Acquaotta, F., Kim, H., Lee, W., Iniguez, C., Forsberg, B., Ragettli, M. S., Guo, Y. L. L., Chen, B. Y., Li, S., Armstrong, B., Aleman, A., Zanobetti, A., Schwartz, J., Dang, T. N., Dung, D. V., Gillett, N., Haines, A., Mengel, M., Huber, V., and Gasparrini, A.: The burden of heat-related mortality attributable to recent human-induced climate change, Nature Climate Change, 11, 492–500, https://doi.org/10.1038/s41558-021-01058-x, publisher: Nature Publishing Group, 2021.
- Webber, R. J., Plotkin, D. A., O'Neill, M. E., Abbot, D. S., and Weare, J.: Practical rare event sampling for extreme mesoscale weather, Chaos: An Interdisciplinary Journal of Nonlinear Science, 29, 053 109, https://doi.org/10.1063/1.5081461, 2019.
  - White, R. H., Anderson, S., Booth, J. F., Braich, G., Draeger, C., Fei, C., Harley, C. D. G., Henderson, S. B., Jakob, M., Lau, C.-A., Mareshet Admasu, L., Narinesingh, V., Rodell, C., Roocroft, E., Weinberger, K. R., and West, G.: The unprecedented Pacific Northwest heatwave of June 2021, Nature Communications, 14, 727, https://doi.org/10.1038/s41467-023-36289-3, 2023.
- Wouters, J. and Bouchet, F.: Rare event computation in deterministic chaotic systems using genealogical particle analysis, Journal of Physics A: Mathematical and Theoretical, 49, 374 002, https://doi.org/10.1088/1751-8113/49/37/374002, publisher: IOP Publishing, 2016.
  - Yiou, P. and Jézéquel, A.: Simulation of extreme heat waves with empirical importance sampling, Geoscientific Model Development, 13, 763–781, https://doi.org/10.5194/gmd-13-763-2020, publisher: Copernicus GmbH, 2020.
- Zeder, J. and Fischer, E. M.: Quantifying the statistical dependence of mid-latitude heatwave intensity and likelihood on prevalent physical drivers and climate change, Advances in Statistical Climatology, Meteorology and Oceanography, 9, 83–102, https://doi.org/10.5194/ascmo-9-83-2023, publisher: Copernicus GmbH, 2023.
  - Zeder, J., Sippel, S., Pasche, O. C., Engelke, S., and Fischer, E. M.: The Effect of a Short Observational Record on the Statistics of Temperature Extremes, Geophysical Research Letters, 50, e2023GL104090, https://doi.org/10.1029/2023GL104090, \_eprint: https://onlinelibrary.wiley.com/doi/pdf/10.1029/2023GL104090, 2023.