# Peer review of "Estimating return periods for extreme events in climate models through Ensemble Boosting"

_EGUsphere, 2025_

## Referee Comment (RC2)

Review – "Estimating return periods for extreme events in climate models through Ensemble Boosting" by Bloin-Wibe et al.

This paper reports a study of a way in which ensemble boosting may be used to estimate the probability of an event more extreme than a reference event by conditioning the boosted ensemble on the occurrence of the reference event. Under suitable conditions, this allows estimates of the probability of the extreme event of interest with a specified level of uncertainty at a lower computational cost than would be possible using a raw, unboosted ensemble. I like this very much because it brings statistical importance sampling concepts to bear on the analysis of storylines, thereby creating the possibility of associating estimates of the probability of recurrence to the cases considered in storyline studies and thus answering a key user question – what are the odds that a damaging event like the one that is considered in the storyline will happen again.

The main issue that the authors struggle with in this paper is how to generate the boosted ensemble in a way that allows them to produce unbiased estimates of the probability of the extreme of interest without incurring excessive computational cost while using a climate model that appears to be relatively expensive to run. (The authors report that the climate model they use is capable of simulating just over 1-year per wall clock day on their computing systems  – which is slow compared to simulation rates, such as 5-years per wall clock day, that would be more amenable to this type of research). The conditions necessary to demonstrate that these estimates are unbiased, which are set out at around line 195 in the paper, essentially require the occurrence of the extreme of interest to be independent of the occurrence of the reference event. This leads to a delicate trade-off in which the authors try to use boosted ensembles where the reference event still has some influence on the occurrence of the extreme of interest while not discernably violating this assumption.

I find the trade-off and the way it is considered a bit unsatisfying because there are elements that seem ad-hoc. Also, the resulting boosted sample of short simulations leads to a sample of extremes the are conditional on the reference event that may have different properties than the extreme of interest. The extreme of interest (such as the maximum 5-day mean temperature observed during the heat dome event) can be thought of as a block maximum where the block is either the Northern Hemisphere summer season (JJA maximum) or, equivalently, a year (annual maximum), because we know now that the event was the annual maximum event in that region. Such block maxima are often assumed to have GEV distributions, based on asymptotic theory and a set of idealized mathematical conditions. In contrast, the sample of maxima obtained from the boosted ensemble are

something like the maxima of 18-day (or so, depending on the length of the boosted runs) blocks, with blocks representative of only part of the summer. The facts that the blocks are shorter alters their statistical properties relative to those of annual or summer maxima, even if after 18-days, the short timescale atmospheric behaviour of the boosted simulations is essentially independent of that of the unboosted simulations. An 18-day block maximum is substantially farther from the "domain of convergence" to the GEV distribution than a 90-day block maximum, particularly in cases when the parent distribution is near Gaussian (convergence to the GEV occurs only very slowly for the block maxima of Gaussian samples). Also, note that we would not expect complete independence from the reference event due to the influence of slower evolving parts of the climate system such as the temperature of the ocean mixed layer, sea and land ice, land surface moisture content, and perhaps stratospheric state.

With these complications in mind, while less computationally efficient, if the question concerns an extreme of an annual maximum temperature index where the annual maximum is presumed to occur sometime in JJA, then a simpler way to proceed would simply be to produce a boosted ensemble of MJJA simulations, perhaps conditioned on some aspect of the lower boundary conditions, such as the initial SST state. This would nevertheless produce an additional 3 annual maxima for the cost of a single additional large ensemble simulation given that the latter must run continuously through full annual cycles. It is recognized, however, that if the complications can be dealt with adequately, then using a strategy based on short "forecasts" (e.g., ensembles of 21-day boosted simulations) would clearly be much preferred.

A second issue that is also considered, but requires further thought, is the selection of the reference event, particularly in cases where the reference event is itself very extreme, as in the 2021 western North America heat dome event. This is because selection bias (e.g., see Miralles and Davison, 2023, https://doi.org/10.1016/j.wace.2023.100584) could have a serious impact on the estimate of the event probability in such cases.

Although the methods used to estimate the conditional probabilities are different, the problem considered in this paper is nevertheless somewhat similar to that of producing probability of precipitation forecasts, as is performed operationally at numerical weather prediction centers as part of each forecast cycle (these forecasts are clearly conditional on assimilated antecedent observations). This suggests that tools used for the verification of short-term probability forecasts could perhaps also be used to test the ensemble boosting approach and thus help determine whether departures from the conditions under which boosting can be used to estimate the probabilities of extreme events impede their interpretation. Note that I've framed this thought in terms of the impacts of departures

from the idealized conditions because it is unlikely that those conditions would ever be satisfied fully, just as the postulates that underpin standard extreme value theory leading to the GEV distribution are never completely satisfied in real applications.

**Some editorial and specific comments** (listed by line number where appropriate)**:**

The overall readability of the paper could be improved. I think the authors would be well advised to further polish the paper with the needs of the reader in mind, which means aiming for text that is somewhat more tutorial that supports reader comprehension as explanations are often very terse. In this same vein, I find the notation a bit awkward, with both superscripts and subscripts denoting time in different contexts. For example, $i$ and $n$ are used to indicate time (as a superscript), $i$ is used to grid row (as a subscript), $t$ is used to indicate time lags (as a subscript) and $T$ means temperature. I don't have good suggestions for simplifying the notation, but complexity of the current notation system impedes reader understanding. The notation system seems a bit non-intuitive and gives the impression that it was the result of a series of quick additions each time something additional was required.

31:     Also cite Philip et al. (2022, https://doi.org/10.5194/esd-13-1689-2022)

50-53: This characterizing of the reliability of inferences based on extreme value theory and approaches that "remain statistical" seems a bit disdainful, particularly given that this paper also relies heavily on statistical concepts.

95:     Overall, I appreciate this well written introduction.

124:    I think this is poorly stated. While the particular interest may be in the upper tail of the distribution of events that could plausibly follow a specified parent event at a given lead time, the boosted ensemble presumably has information about the entire distribution at that lead time conditional on the parent event.

127:    "parent event will grow" → "parent event initially will grow" (as you show later, and as is well known, the magnitude of the errors does not grow forever).

135:    Why did you choose to perturb Q rather than some other quantity?

161-162: It's not clear what would constitute a "good enough sample", particularly given that "good enough" forms the basis for a key assumption.

177:    The assumption that $\mathbb{P}(T > T_{ext})$ is constant throughout the summer is not normally made when using statistical block maximum approaches for estimating upper tail probabilities; I think that assumption should be discussed more thoroughly here.

The presence of the annual cycle implies that daily surface air temperature (including daily values of the daily mean, maximum or minimum) are *not* identically distributed throughout the summer – which is what seems to be assumed here.

179 (eq. 9): Evidently, it is necessary to estimate all three terms on the right well, which has implications for reference event selection from an existing large ensemble and for the size of the boosted ensemble. It would be useful if these implications could be considered.

190: Rather than simply counting events, perhaps other, possibly more efficient methods of estimating these upper tail probabilities could be considered – for example, by computing these probability estimates from fitted extreme value distributions. Doing so would account not just for the frequency of exceedance above some value, but also the form of the distribution of those exceedances.

217: It is unclear which three parameters can be set by the experimenter (see my overarching comment about the notation).

220: I'm not sure I understand what exactly is set to 0.75 or 0.3, nor what that implies for the reference temperature or for the set $AC_t^\varepsilon$. See my overarching comment above – a more tutorial "handholding" approach would be beneficial if you want readers to understand well what this paper proposes.

226: I'm a bit confused by the various N's here and what they represent. It seems to me that the reference climate simulations must run through entire annual cycles over an extended period of time if you are going to simulate the entire variance spectrum (including internal variability associated with slow processes). Such long runs provide one annual extreme per year. Under the right conditions, a boosted 21-day simulation could hopefully provide an additional realization of an annual extreme at only a small fraction of the cost of running through an annual cycle (e.g., 21/365 or <6%). This would represent a speed-up of a factor of ~17 ...

245-247: The 4000-year preindustrial control contains 80 non-overlapping 50-year segments, so if I were to choose two at random, as implied on line 247, it would be unlikely that they end up being exactly adjacent to each other (a simple combinatorial argument demonstrates that the probability of that occurring at random is 1/40 = 0.025). So, exactly how were those two periods selected? Also, how is time referenced – years since the start of the PI control?

272-281: I'm again a bit confused. First, there seems to be a strong implicit assumption that the 30-member large ensemble is indistinguishable from the 100-member ensemble (despite presumably having been produced at different times by different

groups using different computing hardware, etc., with each group learning by making their own mistakes). If they are indistinguishable, then pooling those two large ensembles together to form an even larger ensemble may allow some further improvement in tail probability estimation before boosting. Second, the idea of "detrending" extremes always raises some concerns because extremes are generally not symmetrically distributed. I don't immediately have suggestions for mitigating potential concerns, but I think at least you should flag to readers that there might be concerns with this kind of procedure in some instances.

282-286: I'm also confused about the numbers 7 and 13. Is 13 a typo (could you have meant 31 – highest value of TXx5d simulated across the 30-member ensemble for each of the 31 years considered?). And why boost only 7 of these events? Was that just a pragmatic decision made in the face of a limited computing budget?

282-290: Something else that is not said is to indicate the details of the time series of daily maximum temperatures is used to calculate TXx5d for the PNW case. Is this a time series of spatially averaged values across some region (if so, what region), or a times series of the maximum value observed over some period across all grid points in a region (again, what region), or simply based on temperatures at a selected representative location (if so, what location)?

374-375: This seems rather ad-hoc. I wonder if this could be formalized in some way to allow a more objective assessment of what the trade-off between relative sampling error and lead time should be. That trade-off implies a source of uncertainty for the estimate of unconditional event probability that you ultimately aim to provide. How that uncertainty depends on the rarity of the reference (parent) event would also be good to formalize, if possible.

394-396: How do you explain this? Note that I wouldn't necessarily accept that the estimates from fitted GEV distributions are "naïve". This seems naïvely dismissive in my view. As I mentioned above, there might be reasons to think that a boosted sample of extremes from short simulations may estimate something different from the thing that can be estimated with annual block maxima.

404-405: Suddenly, there is discussion of an event "E1" without a definition of the event. (please improve the notation …).

420: How do we know that it is a more robust estimate, or that it more realistic than other estimates that suggest that the heat dome was a very low probability event?

502-504: I would agree that extrapolation into the deep upper tail of the GEV distribution has many limitations, but the fundamental limitation is not uncertainty

quantification via bootstrapping, as suggested here. Rather, it is the assumption of max-stability (the notion that block-maxima of samples of block-maxima again have GEV distributions with the same shape parameter). This implies that there will be no "surprises" in the unobserved parts of the tail that do not conform with the behaviour seen in the sampled parts of the distribution. I think this implicitly assumes that there is only one physical process that generates extremes at a given location – while in reality, it might be reasonable to think that the PNW heat dome event reflects the impact of a process that is distinct from the process (or collection of processes) that produces the annual maximum temperature event in the region in most years.

---

## Author Comment (AC1)

Egusphere-2025-525: Estimating return periods for extreme events in climate models through Ensemble Boosting

Response to the reviewer's comments by Luna Bloin-Wibe, Robin Noyelle, Vincent Humphrey, Urs Beyerle, Reto Knutti and Erich Fischer.

We would like to thank both reviewers for their efforts in providing constructive comments and suggestions. The text below (in blue) addresses the feedback received (in black), with figure and line numbers referring to the revised manuscript. Text in "quotation marks" refers to the content of the manuscript, with text in *italic* showing additions or changes with respect to the original submission.

**Reviewer 1**

This paper introduces a new framework to estimate return periods of rare climate extremes using ensemble boosting and conditional probability theory. The technique enhances the sampling of extreme events through targeted perturbations, thereby improving return period estimates without requiring prohibitively long control runs. The method is carefully developed and applied to CESM2 under both stationary and transient conditions, including an application to the 2021 Pacific Northwest heatwave.

The manuscript is clearly written and proposes a promising and computationally efficient approach. That said, several assumptions and empirical decisions underlie the method, and their implications for robustness and generalizability are not fully explored. I believe the paper could make a strong contribution after revisions addressing the following points.

We thank the reviewer for the positive feedback, as well as for the suggestions outlined here and detailed in the following paragraphs. We revised the manuscript in accordance with the points raised (see specific responses below).

Main Comments

Assumptions in the estimator: The validity of the boosting estimator depends on assumptions that may deserve further testing or clarification:

- The parent ensemble is assumed to adequately sample the antecedent condition set ($AC^0_t$). Section 4.2 includes some helpful discussion and testing of the number of parent events used, particularly in the pre-industrial slices. However, it would be useful to further clarify what aspects of the full antecedent condition space are critical for representativeness, and whether longer-term variability (e.g., decadal modes) might still be undersampled.
  Thank you for mentioning this important point. Although the contributing physical mechanisms to heatwaves are well understood (anticyclonic conditions leading to higher insolation and adiabatic heating through subsidence, diabatic heating e.g. through antecedent dry soils…, see introduction), we have a more limited understanding of the

mechanisms that are critical for representativeness within parent antecedent conditions. If we knew the function giving the probability to reach an extreme given the current state - the so-called committor function (see Miloshevich et al. (2023)[1]), we would select the most representative parents based on these criteria. Instead, we therefore select based on temperature as a first order approximation, and sample several parents, thus widening the sample and increasing the likelihood of representativeness. The implicit conjecture is that by selecting the most intense heat waves, we expect that the representative antecedent conditions are sampled.

Additionally, the conditions for a representative sample is not guaranteed to be the same over different regions, especially since we know that heatwaves form due to different mechanisms on different latitudes. This is another reason why we choose to sample "blindly".

Figure A6 also shows that estimating return periods of boosted simulations is not very sensitive to removing the boosted simulations of one parent event from the boosted ensemble. This increases confidence that the antecedent conditions are already sufficiently well sampled.

We have highlighted these points by adding the following sentence in the methods (l. 171-176):

*"The exact conditions constituting a representative sample are unknown, since we do not know the function giving the probability to reach an extreme given the current state - the so-called committor function (see Miloshevich et al., 2023). Additionally, if we did, it may not be transferrable across different regions. Instead, temperature at $t\_0 = 0$ is used as a first order approximation to determine the right antecedent conditions (at lead time $t<t\_0$) for a heatwave --- if the antecedent conditions produced a heatwave, it could potentially produce a more extreme one."*

The 50-year time slices are selected to reflect a typical time scale of reliable historical records available in all regions of the world, and could indeed lead to uneven sampling, due to long-term variability as mentioned above. This is one of the reasons postulated as to why test slice 2 doesn't follow the control period as closely as test slice 1 (l. 420-423). This could be made clearer in the text, and we added the following statement in the manuscript in the method section (l. 333-336):

*"The length of the time slice is selected to reflect typical timescales of available historical records. While this can serve as a comparison to historical extreme event attribution studies, this time scale might present highly uncertain estimations of $T\_ref$, in particular due to the limited sample and the long term temperature variability effects."*

Could the estimator be biased or unstable if AC^0_t is only sparsely or unevenly populated?

The estimator remains unbiased regardless of the sample as long as the conditions stated in l. 204-209 are fulfilled. Whether AC^0_t is well sampled or not will affect the precision of the estimation and its relative errors rather than whether it is biased or not. The choice of lead time, however, could lead to a biased estimator, as discussed in

- The method also assumes independence between $\hat{p}_{T \ge T_{\text{ref}}}$ and the conditional ratio $\frac{\hat{p}_{T \ge T_{\text{ext}}} \mid AC^{\epsilon_t}}{\hat{p}_{T \ge T_{\text{ref}}} \mid AC^{\epsilon_t}}$. This assumption is discussed in Appendix A1 and appears plausible in the authors' setup, but it is not directly tested. It would be helpful to evaluate how return period estimates change when varying $T_{\text{ref}}$ or the number of parent events, to better understand whether this independence holds in practice.
  It would indeed be insightful to test the effects of the number of parents/value of $T\_ref$. However, since ensemble boosting is costly, our options for validation and experimenting are limited. Therefore, we indirectly test for this effect in Figure 8 and Appendix Figure A5 by looking at the correlation between $T\_ref$ and the 90th percentile of the boosted ensemble (see also discussion section 4.3). Additionally, we also look at the effect of removing one parent in Figures 9 and A7. Finally, following the comment on validation in a simpler setting below, we will evaluate the effect of the number of parent events chosen in an AR1 model, where we have less computational constraints.

Methodological choices and tuning: Several aspects of the boosting design are empirical and would benefit from more context or testing:

- The use of specific humidity as the perturbation variable is described as effective, but it's not entirely clear why this variable was chosen over others. Were other variables (e.g., temperature, geopotential height) tried? If so, the rationale could be made more explicit.
  Since, as mentioned by the reviewer below, the perturbation stays within numerical noise limits, we do not expect the atmospheric variable chosen to make a difference. Both Gessner et al. (2023)[2] and Fischer et al. (2023)[3] use specific humidity when performing boosting, while Gessner et al. (2021)[4] and Gessner et al. (2022)[5] use temperature, with similar results. The following sentence was added in the manuscript (l. 145-148) in order to clarify this:
  *"While a systematic test has not been implemented, we do not expect the choice of variable to influence results, since the perturbation stays within numerical noise limits. Both specific humidity and temperature have been used in previous studies, with comparable results (\cite{gessner_very_2021}, \cite{gessner_multi-year_2022}, \cite{gessner_developing_2023}, \cite{fischer_storylines_2023})."*

- The perturbation magnitude $(1 + 10{-13} \cdot R 10^{-13} \cdot R 10{-13} \cdot R)$ seems designed to stay within numerical noise limits. Still, it would be good to mention whether other values were tested, or whether results depend on this factor at all.
  We follow the setup outlined in Fischer et al. (2023)[3], but previous comparable studies have performed boosting with perturbations of *$10^{-18}$ (Gessner et al., 2023)[2] and $10^{-14}$ (Gessner et al., 2021,2022)[4,5]. These papers are now cited in the added sentence above.

- The lead time of −12 days is said to balance realism and divergence. Section 4.3 justifies this choice based on ensemble spread and trajectory divergence, which is useful. Still, have return period estimates themselves been tested for robustness to this choice?
  In Figure 1, we show three different lead times, but in the paper in general, all figures that don't explicitly show difference between lead time (Figures 8,9, A5 and A7), use pooled lead times from -18 to -13 days, rather than only -12 days. Figures 4, A2, A3, A4 and A6 show return period calculations as a function of lead time (either pooled or standalone).

  Pooling across lead times (as shown in Fig. 4d) seems helpful — if that's generally recommended, it might be worth saying so directly.
  Yes, we would generally recommend to proceed as such. This is mentioned in discussion section 4.3, and is highlighted in the second bullet point of the conclusion (l. 664-673)

This isn't to criticize the empirical design — that's often necessary in early-stage methods — but documenting what was tested and what was fixed would strengthen the work and help future applications.

Validation in a simpler, fully controlled setting

To me, one of the most convincing ways to build confidence in the proposed estimator would be to test it in a much simpler, controlled setting — for example, a low-order stochastic model or linear inverse model where the true return periods are known (or can be computed empirically over very large samples).

This would allow a direct comparison between the boosted estimator and ground truth, and help isolate where biases or over-/under-confidence may arise. It could also help evaluate how the estimator behaves when assumptions like conditional independence or adequate AC□ sampling are or aren't satisfied.

Even a basic demonstration of this kind would be extremely informative and, in my view, would strengthen the paper considerably.

We thank the reviewer for this important and valuable suggestion. We have implemented an autoregressive (AR1) model in order to better evaluate the different assumptions underlying the paper.  In the interest of space, the entire addition will not be rendered here, but can be found in the revised manuscript (section 2.3, Validation with auto-regressive model). Here, we show the added figures, and the main interpretation:

[Figure]

Temporal evolution of parent and boosted simulations

[revised manuscript text omitted]

*error calculation (see Section \ref{sec:th_rel_err}), this reduction of uncertainty is more pronounced for a larger N_batch (Panels d,e}) or N_parent (Panels c,e), with the smallest uncertainty range found in Panel e, where the total computational cost is largest (10 000 simulations). This highlights that while sparse sampling of either the parent or the boosted ensemble can increase uncertainty and relative error with respect to the ground truth, it does not bias the return period estimation.*

*Finally, comparing Panels c and d allows for evaluating how N_parent and N_batch affect the boosted simulations and their subsequent return period estimation. At the same computational cost as Panel c, Panel d presents boosted simulations with higher return levels and similar, or at times smaller, levels of uncertainty. This is because P_Tref here is much higher (0.01 compared to 0.1 for Panel c), thus increasing the chances of events in the boosted ensemble with larger return periods.*

*By exploring different configurations of the relevant boosting parameters N_parent and N_batch, this evaluation thus strengthens confidence in estimating return periods using the boosting estimator. This confidence is underscored by a more extensive uncertainty sampling of both the parent and boosted ensemble than what would be possible in a fully-coupled climate model. However, it is important to note that the Ornstein-Uhlenbeck process is one-dimensional, while a state-of-the art climate model has millions of degrees of freedom. Therefore, the transferability of results found here are limited, in particular for the size of the parent ensemble necessary to sample AC^e_t well."*

Confidence interval handling
The method appears to yield narrower confidence intervals than GEV-based estimates in some cases. While this could reflect improved sampling, it might also result from underestimating uncertainty in the boosted setting. Appendix A mentions that bootstrapping is used, which is helpful. Still, it would be good to clarify whether the intervals fully reflect all sources of uncertainty (e.g., finite Nparent, dependence structures, or sensitivity to NbN_bNb).
We thank the reviewer for this suggestion. The confidence intervals are larger for the GEV-based estimates of test slice 1 (sample size =50) than for the boosted simulations (sample size 3000), but that primarily comes down to sample size. Compared to the confidence interval of the control period (sample size 4000), the confidence intervals are much larger in the boosted simulations.

Since we purely use bootstrapping to calculate the confidence intervals, they do not reflect, as is mentioned by the reviewer, sources of uncertainty related to experimental design such as the number of parents used (the confidence interval around the test slices comes from bootstrapping the test slice itself, not the control simulation, for example). This is now highlighted with the following sentence in the method section (l. 367-369):
*"Note that since we do not change the experimental sample when calculating the confidence interval (e.g. selecting and boosting a different parent ensemble), this confidence interval may underestimate certain sources of uncertainty such as slow modes of variability of the climate system"*

Minor comments/suggestions

Nonstationarity correction. Line 279: The paper states that results are corrected for non-stationarity, but the method used for that correction isn't described in much detail. How is the rolling climatology computed? Is it applied to each member individually or to ensemble means? And does the choice of window matter?

A more detailed description of the non-stationarity has been added (l. 376-379), as follows: "*This is done by linearly de-trending the TXx5d time series: for each year and each member of the Large Ensemble, a 100-member Large Ensemble of the same model \citep{rodgers_ubiquity_2021} is used to produce a day-of-year mean across members and a three-year window (the year in question, and 1 year before and after), around each year. The three-year window was chosen to create a larger sample of similar years, without adding the climate change signal present over longer time scales.*"

Section 2.3: Including computational cost (e.g., node-hours or wall-clock time) for the boosted ensemble would help support the method's efficiency claims.

We thank the reviewer for this useful suggestion. A discussion of the cost of producing parent members and boosted members can be found in section 4.1. However, computational cost is only given in simulation days. Additionally, in l.221, we mention the output capability of our computational system as "simulating approximately 1.1 climate model years in 24 hours", which does not specify whether it is wall-clock hours or node hours. We therefore changed this line to "simulating approximately 1.1 climate model years in 24 *node* hours, *with a capacity of up to 4 nodes*". Similarly, we changed " this approach would take approximately 1.5 years" to "this approach would take approximately 1.5 *node* years".

Additionally, we added an appendix figure to visually differentiate the distributions of the parent and boosted members (Appendix Figure 1, seen in reviewer 2's comment about the original manuscript's l.124). This will visually show the efficiency gain, while responding to suggestions from Reviewer 2.

Notation: Several variables (e.g., TXx5d, $T_b^n$TbnT_b^nTbn, $T_{\text{ext}}$TextT_{\text{ext}}Text) appear. A glossary or symbol table might help readers.

We thank the reviewer for this suggestion! A glossary has been added at the beginning of the method section.

Confidence intervals: Have you tested how return period confidence intervals behave if $N_b = 1500$Nb=1500N_b = 1500Nb=1500 or 6000? Even a brief comment would help.

With the simple AR1 model, we can directly assess this effect. As can be seen in the newly added Figure 4, confidence intervals do get slimmer as $N_b$ increases (either through $N_{parent}$ or $N_{batch}$), but these intervals stay similarly slim or slimmer than the confidence interval of the parent ensemble.

Alternative thresholds: Appendix A briefly discusses threshold sensitivity, but the main text might benefit from a more explicit statement. Would estimates change significantly if parents are

selected above the 95th or 99th percentile instead of 90th?

Changing the threshold of selection is equivalent to either changing T_ref and keeping the sample size equal (which means changing the number of parents) or keeping T_ref but changing the sample size. The former – evaluating the effect of the number of parents selected – is addressed in a comment above (first bullet point of page 3), and the latter is discussed theoretically in section 2.2, and is additionally evaluated in our added section 2.3 on the AR1 model (see "Validation in a simpler, fully controlled setting").

It would be helpful for readers to clarify these equivalences. Therefore, we have added the following sentence in the manuscript (l. 344-347), when describing the experimental setup of choosing T_ref = 0.1: *"This value is chosen to balance rareness in the Parent Ensemble, and the ability to robustly estimate P_Tref in the test slice. Changing the threshold of selection is equivalent to either changing T_ref and keeping the sample size equal (which means changing the number of parents) or keeping T_ref but changing the sample size. The former will be evaluated in the discussion section, and the latter in the following subsection."*

This is a creative and carefully implemented study with a potentially valuable method for return period estimation. The framework is promising and the examples are well chosen. I appreciate that the authors are transparent about the method's limitations, particularly regarding subjective choices and empirical design. That said, several of these choices and assumptions could still benefit from additional testing and sensitivity analysis. In particular, validating the method in a simple, controlled setting where return periods can be measured directly would provide a powerful test of its performance. With these revisions, the paper would be a strong contribution to the literature on climate extremes.

Cristian Martinez-Villalobos

**Reviewer 2**

This paper reports a study of a way in which ensemble boosting may be used to estimate the probability of an event more extreme than a reference event by conditioning the boosted ensemble on the occurrence of the reference event. Under suitable conditions, this allows estimates of the probability of the extreme event of interest with a specified level of uncertainty at a lower computational cost than would be possible using a raw, unboosted ensemble. I like this very much because it brings statistical importance sampling concepts to bear on the analysis of storylines, thereby creating the possibility of associating estimates of the probability of recurrence to the cases considered in storyline studies and thus answering a key user question – what are the odds that a damaging event like the one that is considered in the storyline will happen again.

We thank the reviewer for their positive feedback.

The main issue that the authors struggle with in this paper is how to generate the boosted ensemble in a way that allows them to produce unbiased estimates of the probability of the extreme of interest without incurring excessive computational cost while using a climate model that appears to be relatively expensive to run. (The authors report that the climate model they use is capable of simulating just over 1-year per wall clock day on their computing systems – which is slow compared to simulation rates, such as 5-years per wall clock day, that would be more amenable to this type of research).

Producing unbiased estimates is indeed a challenging crux of this paper. We thank the reviewer for remarking on the computational expenses, since the current statement in the paper is vague. What is referred to is the node hours, i.e. the number of years simulated in a day per node. Since we have a computational capacity of up to 4 nodes, we have amended the sentence (l. 221) in the following way: "simulating approximately 1.1 climate model years in 24 *node* hours, *with a capacity of up to 4 nodes*".
However, the efficiency gain is stated in relative terms, and would therefore be substantial for any other climate model used, even if it would be cheaper to run.

The conditions necessary to demonstrate that these estimates are unbiased, which are set out at around line 195 in the paper, essentially require the occurrence of the extreme of interest to be independent of the occurrence of the reference event. This leads to a delicate trade-off in which the authors try to use boosted ensembles where the reference event still has some influence on the occurrence of the extreme of interest while not discernibly violating this assumption.
The conditions for unbiasedness stated in the methods do not exactly require the reference and boosted extreme events to occur independently of one another, but rather that: (1) the probability of exceeding the reference temperature T_ref is independent of how much more likely it is to exceed the temperature of a boosted simulation T_ext than T_ref in the boosted ensemble; and (2) the probability that a boosted simulation exceeds the temperatures T_ref within the boosted ensemble (conditional on boosting) is independent from the probability of any other boosted simulation exceeding T_ref (likewise for T_ext). This nuance is important: if T_ref and T_ext were independent, the boosting would not serve its purpose, which is resampling simulations conditional on an extreme.
This has been further clarified by adding the following statement in the methods section (l. 210-212): *"Note that these assumptions do not require that T_ref and T_ext occur independently of one another, but rather explore the relationship between T_ref and T_ext within the boosted ensemble, and their link to T_ref."*

I find the trade-off and the way it is considered a bit unsatisfying because there are elements that seem ad-hoc. Also, the resulting boosted sample of short simulations leads to a sample of extremes that are conditional on the reference event that may have different properties than the extreme of interest.

The boosted simulations will indeed be conditional on the same antecedent conditions as their parent. This is a desired effect: through this conditionality, we expect them to produce heatwaves, especially ones that are more extreme than their parent.
Estimations of probability through frequency of occurrence, directly applied to the boosted simulations, will therefore only give conditional probabilities. However, the boosting estimator is designed such that we can calculate unconditional probabilities of any boosted simulation that exceeds T_ref. This is done by linking the boosted and parent simulations by what they have in common, which are the antecedent conditions that could lead to an exceedance of T_ref. Sampling these antecedent conditions well, as mentioned by both reviewers, is therefore crucial to decrease the variance of the boosting estimator. We argue that our results are satisfactory enough to build confidence in the estimator: First from a theoretical and simplified point of view in Section 2.2 (theoretical relative errors) and the newly added Section 2.3 on validation with an Ornstein-Uhlenbeck process, and then, with a fully-coupled climate model, in Section 3.2 and the Discussion section.

The extreme of interest (such as the maximum 5- day mean temperature observed during the heat dome event) can be thought of as a block maximum where the block is either the Northern Hemisphere summer season (JJA maximum) or, equivalently, a year (annual maximum), because we know now that the event was the annual maximum event in that region.
Such block maxima are often assumed to have GEV distributions, based on asymptotic theory and a set of idealized mathematical conditions. In contrast, the sample of maxima obtained from the boosted ensemble are something like the maxima of 18-day (or so, depending on the length of the boosted runs) blocks, with blocks representative of only part of the summer. The fact that the blocks are shorter alters their statistical properties relative to those of annual or summer maxima, even if after 18-days, the short timescale atmospheric behaviour of the boosted simulations is essentially independent of that of the unboosted simulations. An 18-day block maximum is substantially farther from the "domain of convergence" to the GEV distribution than a 90-day block maximum, particularly in cases when the parent distribution is near Gaussian (convergence to the GEV occurs only very slowly for the block maxima of Gaussian samples). We thank the reviewer for highlighting this important condition for using GEV distributions. As a clarification, since we here work with summer temperature anomalies, the JJA maximum is not completely equivalent to the annual maximum. This has been made clearer by adding a glossary at the beginning of the method section of the paper.
While we do not fit GEV distributions directly to the boosting estimator, the blocks of boosted simulations do indeed need to fulfill the same conditions as the parent ensemble to be able to compare them directly.
However, since we only calculate return periods for boosted simulations that exceed T_ref (the ones that do not exceed it are only accounted for indirectly when we calculate the probabilities of T_ref and T_ext conditional on boosting), in reality these are not 21-day maxima (even if they are only run for 21 days). Firstly, T_ref is already a rare summer maximum, which means that the boosted simulations that exceed it are likely to also be the summer maximum. We tested this in our PI-control samples (test slice 1 and 2) and found that for the simulations that do exceed T_ref, only 4.2 (T2) - 4.6 (T1)% have a higher temperature in the following 60 days. We

decided against keeping these events in the sample, since they occurred substantially later than the antecedent conditions we sampled on.

Secondly, since the boosted simulation shares the trajectory of its parent until it is perturbed, the boosted simulations are longer than just 21 days. However, only summer days are considered when looking at the block maximum, so the length of the boosted simulation depends on when in the summer the parent heatwave was perturbed. For example, perturbing a parent simulation on 10 August and running it for 21 days would yield a full summer maximum, while perturbing on 1 June would yield a 21-day maximum.

Since this has not been discussed in the paper in detail, we added the following statement (l. 356-362):

*"In order to directly compare return periods calculated using the boosting estimator, the TXx5d need to reasonably fulfill the domain convergence conditions of EVT theory. While 21 days is not as long as the approximately 100 days of a full summer, we postulate that the boosted simulations comply reasonably well with GEV requirements for the following reasons: Firstly, since T_ref is already a rare summer maximum, the boosted simulations that exceed it are likely to also be the summer maximum. Indeed, for the simulations that do exceed T_ref, only 4-5% have a higher temperature in the following 60 days. Secondly, the parent heatwaves do not all occur in the beginning of the summer. Since the boosted simulation shares the trajectory of its parent until it is perturbed, the boosted simulations are generally longer than just 21 days."*

Also, note that we would not expect complete independence from the reference event due to the influence of slower evolving parts of the climate system such as the temperature of the ocean mixed layer, sea and land ice, land surface moisture content, and perhaps stratospheric state.

As mentioned above, we do not expect, nor want, complete independence from the reference event. However, we do want the boosted simulations to exceed T_ref or T_ext independently of one another. Choosing more parents, which are independent from each other, to be boosted, and increasing the lead time, increases this independence.

With these complications in mind, while less computationally efficient, if the question concerns an extreme of an annual maximum temperature index where the annual maximum is presumed to occur sometime in JJA, then a simpler way to proceed would simply be to produce a boosted ensemble of MJJA simulations, perhaps conditioned on some aspect of the lower boundary conditions, such as the initial SST state. This would nevertheless produce an additional 3 annual maxima for the cost of a single additional large ensemble simulation given that the latter must run continuously through full annual cycles.

We thank the reviewer for this observation - the procedure above would be conceptually simpler. Indeed, when calculating computational efficiency, we compare the boosting method to the cost of just generating MJJA simulations (see Section 4.1). Here, we find that the boosting estimator can use approximately 6 times less computational resources and yield less erroneous results for a specific extreme of interest than when generating MJJA simulations.

It is recognized, however, that if the complications can be dealt with adequately, then using a

strategy based on short "forecasts" (e.g., ensembles of 21-day boosted simulations) would clearly be much preferred.

A second issue that is also considered, but requires further thought, is the selection of the reference event, particularly in cases where the reference event is itself very extreme, as in the 2021 western North America heat dome event. This is because selection bias (e.g., see Miralles and Davison, 2023, https://doi.org/10.1016/j.wace.2023.100584) could have a serious impact on the estimate of the event probability in such cases.

The choice of the Pacific North-West region is indeed because of the 2021 heatwave, but we only use climate model data when estimating its probability. This means that the event itself is not included (since it does not exist in the model), and would therefore not bias the estimation, in particular since we do not use GEV distributions with the boosting estimator.
It is, however,  important to sample the antecedent conditions well when selecting the parent (or reference) events. Therefore, we generally recommend a larger sample of parent events (see discussion section 4.2 and conclusion). Additionally, the typicality argument, postulating that the more extreme a heatwave is, the more it is typical (or similar to other events of that magnitude), gives some confidence that the number of parents necessary for a good enough sample may not need to be too large.

Although the methods used to estimate the conditional probabilities are different, the problem considered in this paper is nevertheless somewhat similar to that of producing probability of precipitation forecasts, as is performed operationally at numerical weather prediction centers as part of each forecast cycle (these forecasts are clearly conditional on assimilated antecedent observations). This suggests that tools used for the verification of short-term probability forecasts could perhaps also be used to test the ensemble boosting approach and thus help determine whether departures from the conditions under which boosting can be used to estimate the probabilities of extreme events impede their interpretation. Note that I've framed this thought in terms of the impacts of departures from the idealized conditions because it is unlikely that those conditions would ever be satisfied fully, just as the postulates that underpin standard extreme value theory leading to the GEV distribution are never completely satisfied in real applications.

We thank the reviewer for this suggestion. However, as far as we understand probability forecast verification tools, they would need the verification to come from the actual resulting weather. Since this does not exist in the climate model context, it would be hard to harness the full potential of such tools.

Some editorial and specific comments (listed by line number where appropriate):

The overall readability of the paper could be improved. I think the authors would be well advised to further polish the paper with the needs of the reader in mind, which means aiming for text that is somewhat more tutorial that supports reader comprehension as explanations are often very terse.

Thank you for this suggestion. Since the paper is quite technical, it is indeed important to emphasize clarity and reader comprehension. In addition to the specific suggestions pointed out

below, a glossary has been added to the beginning of the method section, as well as several smaller readability changes.

In this same vein, I find the notation a bit awkward, with both superscripts and subscripts denoting time in different contexts. For example, i and n are used to indicate time (as a superscript), i is used to grid row (as a subscript), t is used to indicate time lags (as a subscript) and T means temperature. I don't have good suggestions for simplifying the notation, but complexity of the current notation system impedes reader understanding. The notation system seems a bit non-intuitive and gives the impression that it was the result of a series of quick additions each time something additional was required.

We thank the reviewer for bringing this to our attention. As mentioned above, a glossary has been inserted for a more clear overview of the symbols used.
It is true that $i$ is used both to indicate different simulations in $T^i$, and to indicate a spatial coordinate in $Q_{ijk}$ and $R_{ijk}$. To differentiate $i$ super- and subscript, we have done the following exchanges: $T := \{T^i \mid i = 1,2...N\}$ becomes $T := \{T^n \mid n = 1,2...N\}$, and $T_b := \{T^n_b \mid n = 1,2...N_b\}$ becomes $T_b := \{T^m_b \mid m = 1,2...N_b\}$.
While this may not be clear from the manuscript in its current form, superscripts are not used to indicate time: for $T^n$, $n$ is used to indicate different simulations, going from 1 to N. In a similar vein, in $T^m_b$, $m$ is used to indicate different boosted simulations, going from 1 to $N_b$.
We have made this more clear in the text when it is introduced (l. 123: "forming a set of temperatures $T := \{T^n \mid n = 1,2...N\}$, *with n indicating the different simulations.*").
Other than the antecedent conditions indicating their perturbation as a superscript, there aren't any other superscripts used in the manuscript.
Finally, since using $t$ for time and $T$ for temperature is a common practice in the field, we will keep this as is.

- 31: Also cite Philip et al. (2022, https://doi.org/10.5194/esd-13-1689-2022)
  The citation is added as suggested.

- 50-53: This characterizing of the reliability of inferences based on extreme value theory and approaches that "remain statistical" seems a bit disdainful, particularly given that this paper also relies heavily on statistical concepts.
  We thank the reviewer for this observation. What is meant in the text is that the difference between GEV extrapolation for return periods and the ensemble boosting approach is that in the latter case, a physically consistent climate model simulation for the extreme is still obtained, which can be analyzed for drivers beyond temperature (Z500, soil moisture…). In the former approach, the physical simulations are limited to the sample at hand. However, we have removed the phrasing "this framework alone remains statistical" in the revised manuscript for clarity and to avoid confusion.

- 95: Overall, I appreciate this well written introduction.
  We thank the reviewer for their positive comment.

- 124: I think this is poorly stated. While the particular interest may be in the upper tail of the distribution of events that could plausibly follow a specified parent event at a given lead time, the boosted ensemble presumably has information about the entire distribution at that lead time conditional on the parent event.

  The boosted ensemble stems from a random perturbation ahead of the peak temperature anomaly of its parent, which leads to a sample of the distribution of the climate state at that time step. However, this line is meant to indicate that we need to perturb ahead of the peak temperature anomaly of the parent, since the goal of the boosting procedure is to harness the antecedent conditions leading to a heat wave in order to get an even more extreme heat wave. The distribution sample (several members) are a side effect of the procedure, since we cannot know which perturbation will lead to the most extreme heat wave.

  To visually represent the difference between the climatological distribution and the distribution of boosted simulations, we have added the following appendix figure to the manuscript:

[Figure]

*Appendix Figure 1: Density distribution of boosted yearly summer maximum of daily maximum temperature anomalies with a running mean of 5 days (TXx5d) compared to that of the climatology. Climatological distribution (in black) is derived from the control period, while the distribution of boosted simulations at lead time -12 (in orange) comes from one parent event. The magnitude of the parent event is highlighted (in blue).*

*Additionally, we have added this description in l. 133-134: "The difference between the climatological distribution and the distribution of the boosted simulations at this lead time is shown in Appendix Figure 1."*

- 127: "parent event will grow" → "parent event initially will grow" (as you show later, and as is well known, the magnitude of the errors does not grow forever).
  This is a good point. The line has been replaced as suggested.

- 135: Why did you choose to perturb Q rather than some other quantity?
  The perturbation is only meant to induce a butterfly effect in the atmosphere, and the atmospheric variable chosen does not make a difference. Please also see our answer to reviewer 1 for a more in depth discussion (Methodological choices and tuning, second bullet point, p.3).

- 161-162: It's not clear what would constitute a "good enough sample", particularly given that "good enough" forms the basis for a key assumption.
  Unfortunately, we do not know what is critical of representativeness within parent antecedent conditions - if we knew, we would select the most representative parents based on these criteria. Instead, we therefore select based on temperature, as a first order approximation. See answer to reviewer 1 for a more in depth answer (first bullet point, p.1).

- 177: The assumption that $\mathbb{P}(T > Text)$ is constant throughout the summer is not normally made when using statistical block maximum approaches for estimating upper tail probabilities; I think that assumption should be discussed more thoroughly here. The presence of the annual cycle implies that daily surface air temperature (including daily values of the daily mean, maximum or minimum) are not identically distributed throughout the summer – which is what seems to be assumed here.
  We thank the reviewer for this suggestion. However, the T in question is not absolute maximum temperature, but yearly summer maximum of daily maximum temperature anomalies with a running mean of 5 days (see l. 121-123), which does not have a seasonal cycle. That being said, since we do not work with standardized anomalies, Figure Z (below) shows the climatology of 5-day running mean summer mean, +- one standard deviation over the PNW region (derived from the 4000-year control simulation). Here, we see that the standard deviation also does not change notably, and the assumption that P(T> T_ext) can be said to hold for the purpose of this study.

[Figure]

Figure X (not in manuscript): Tx5 anomaly mean (in black), with +- 1 standard deviation in gray.

- 179 (eq. 9): Evidently, it is necessary to estimate all three terms on the right well, which has implications for reference event selection from an existing large ensemble and for the size of the boosted ensemble. It would be useful if these implications could be considered.
The relative error of the boosting estimator does indeed depend on these three terms, and this is analyzed theoretically in section 2.2, where we see how the size of the original and boosted ensemble affects theoretical relative error. Additionally, in section 4.2, we discuss the effect of the number of parents selected (which, in a fixed sample size, affects the reference temperature threshold), in particular that the boosting estimator will always have errors at least that of P(T_ref). Finally, in the added evaluation section, we analyze the output of a simple autoregressive model, where we can directly look at results when varying the number of parents selected and the total boosted ensemble size.

- 190: Rather than simply counting events, perhaps other, possibly more efficient methods of estimating these upper tail probabilities could be considered – for example, by computing these probability estimates from fitted extreme value distributions. Doing so would account not just for the frequency of exceedance above some value, but also the form of the distribution of those exceedances.
We thank the reviewer for this suggestion. However, we did not include GEV estimations in the boosting estimator for the following reasons:
- P(T>T_ref): in Figure 5 and 9, we see that the naively estimated return periods of T_ref (blue diamonds) are similar to what we find with the median GEV fit, with the boosted and GEV confidence intervals also being similar at that return level. Additionally, we know that GEV estimates systematically bias return times for small samples[6]. Therefore, we therefore choose the naive estimator.
- P(T>=T_ext | AC) and P(T>=T_ext | AC) are conditional on parents in AC, which

means they would not follow a typical block maxima distribution. Therefore, we cannot fit their distribution to a GEV law.

- 217: It is unclear which three parameters can be set by the experimenter (see my overarching comment about the notation).
An explanation of which parameters can be set by the experimenter has been added to the manuscript:
"while the first three terms, *N, N_b and P_Tref,* are parameters that can be set by the experimenter"

- 220: I'm not sure I understand what exactly is set to 0.75 or 0.3, nor what that implies for the reference temperature or for the set AC^e_t. See my overarching comment above – a more tutorial "handholding" approach would be beneficial if you want readers to understand well what this paper proposes.
We thank the reviewer for pointing out how this paragraph can be improved. The sentence has been changed to:
"For illustration purposes, *we choose values of 0.75 and 0.3 for the estimation of P_(Tref|AC)$, which amounts to cases where either 75% or 30% of the boosted simulations exceed T_ref. These values were chosen because they* correspond to the typical value we find in practice (see Results section 3.2)."

- 226: I'm a bit confused by the various N's here and what they represent. It seems to me that the reference climate simulations must run through entire annual cycles over an extended period of time if you are going to simulate the entire variance spectrum (including internal variability associated with slow processes). Such long runs provide one annual extreme per year. Under the right conditions, a boosted 21-day simulation could hopefully provide an additional realization of an annual extreme at only a small fraction of the cost of running through an annual cycle (e.g., 21/365 or <6%). This would represent a speed-up of a factor of ~17 …
We thank the reviewer for highlighting that the explanation stated in the current version of the manuscript needs to be clearer. When comparing costs, we decided that the fairest comparison would be with MJJA ensemble runs (roughly 100 days), rather than full years, since we are only analyzing summer anomaly maxima. However, before being able to generate boosted simulations, one also needs to run a small sample of simulations, from which one selects the parent ensemble. Therefore, the total cost of boosting should take into account these simulations as well. That is why their total cost is N + N_b = 50*100 + 500*21 days, for a parent ensemble of 50 years (run for 100 days each), and 500 boosted simulations (run for 21 days each). To compare this value to an equally expensive non-boosted sample, we need to convert the 500 simulations of 21 days to X simulations of 100 days (X=105 here).
To clarify this, in addition to the glossary added at the beginning of the methods section, we have changed the paragraph as follows:
"Here, three different configurations of the boosting estimator are compared to the errors of the naive estimator. *Since we need to run a sample of non-boosted parents before*

*generating boosted samples, the total cost of the boosted simulations takes this into account ($N+N_b$ = 50\*100+500\*21 days, for a parent ensemble of size 50, where each simulation is run for 100 days, and a boosted ensemble of size 500, where each simulation is run for 21 days). To directly compare these results with the naive estimator, we generate a non-boosted sample where N is equivalent, in terms of computational resources,* to those of each boosting configuration. For example, N=*105\*100 days* is equivalent to *$N+N_b$=50\*100+500\*21 days*, since a boosted simulation only needs to be run for 21 days, while the reference climate model simulation needs approximately 100 days to generate a full summer."

- 245-247: The 4000-year preindustrial control contains 80 non-overlapping 50-year segments, so if I were to choose two at random, as implied on line 247, it would be unlikely that they end up being exactly adjacent to each other (a simple combinatorial argument demonstrates that the probability of that occurring at random is 1/40 = 0.025). So, exactly how were those two periods selected? Also, how is time referenced – years since the start of the PI control?

  We thank the reviewer for pointing this out. The exact timing of the first period was chosen at random, while the second was indeed chosen to be adjacent to the first, to be able to form a third, 100-year test slice (see appendix figure A4). It is acknowledged that this setup is not optimal to sample long term natural variability, however, since it means that the two test slices could present some effects of long term variability not necessarily present in two random test slices. Nevertheless, due to the computational costs of ensemble boosting, this "legacy" setup cannot be changed.
  The years are indeed referenced as the time since the start of the PI control.
  We have made both of these points clear in the manuscript by changing the original phrasing in l. 329-331 to:
  *"While these ranges are adjacent to one another, the starting point of the total time range is* selected randomly within the control run. *Additionally, since the time is only referenced as time since the start of the simulation,* the years do not bear any meaning or relation to real-world weather at that time.*"*

- 272-281: I'm again a bit confused. First, there seems to be a strong implicit assumption that the 30-member large ensemble is indistinguishable from the 100-member ensemble (despite presumably having been produced at different times by different groups using different computing hardware, etc., with each group learning by making their own mistakes). If they are indistinguishable, then pooling those two large ensembles together to form an even larger ensemble may allow some further improvement in tail probability estimation before boosting. Second, the idea of "detrending" extremes always raises some concerns because extremes are generally not symmetrically distributed. I don't immediately have suggestions for mitigating potential concerns, but I think at least you should flag to readers that there might be concerns with this kind of procedure in some instances.

  We thank the reviewer for these suggestions. The reason for distinguishing the two Large Ensembles does indeed come from a technical point of view, and could be better

explained from our side. We have added the following explanations when introducing the 100-member Large Ensembles in the method section:

"*However, this publicly available data set is not locally bit-by-bit reproducible, which is necessary to generate boosted simulations. It will therefore act as a separate, larger data set that* can provide return periods that are more precise than those of the 30-member Large Ensemble, since here, N=100 * 31 = 3100."

Detrending extremes does indeed come with caveats, but since it is performed by removing the ensemble mean, we do not expect artifacts in the distributional tail. For clarity, we have highlighted this by adding a more detailed description of the de-trending methodology (see the first point in the section "Minor comments/suggestions").

- 282-286: I'm also confused about the numbers 7 and 13. Is 13 a typo (could you have meant 31 – highest value of TXx5d simulated across the 30-member ensemble for each of the 31 years considered?). And why boost only 7 of these events? Was that just a pragmatic decision made in the face of a limited computing budget?

  13 is not a typo, but it is acknowledged not to be an intuitive number to use. Since we use the same boosting setup and parents as a previous study [3], where no stationarity correction was performed, the absolute ranking of the top events with our metric and setup is 7 out of 13. This is explained in l 386-390. Additionally, only 7 events were boosted, to be able to generate enough boosted simulations of each parent (in the previous study, larger boosted ensembles were prioritized over the number of parents, while this current study emphasizes the recommendation for future projects to include more parents).

- 282-290: Something else that is not said is to indicate the details of the time series of daily maximum temperatures is used to calculate TXx5d for the PNW case. Is this a time series of spatially averaged values across some region (if so, what region), or a times series of the maximum value observed over some period across all grid points in a region (again, what region), or simply based on temperatures at a selected representative location (if so, what location)?

  This is partly already detailed in lines 323-326: "We seek to estimate the return levels of the yearly maximum of 5-day rolling average of daily maximum temperature anomalies (TXx5d) for a region corresponding to the PNW heatwave (45–52$\circ$N, 119–123$\circ$W), corresponding to the region used by Fischer et al. (2023)*."*

  For clarity, we have changed it to "We seek to estimate the return levels of the yearly maximum of 5-day rolling average of daily maximum temperature anomalies (TXx5d), *spatially averaged over the region of* the PNW heatwave (45–52$\circ$N, 119–123$\circ$W), corresponding to the region used by Fischer et al. (2023).*"*

- 374-375: This seems rather ad-hoc. I wonder if this could be formalized in some way to allow a more objective assessment of what the trade-off between relative sampling error and lead time should be. That trade-off implies a source of uncertainty for the estimate of unconditional event probability that you ultimately aim to provide. How that uncertainty

depends on the rarity of the reference (parent) event would also be good to formalize, if possible.

Thank you for suggesting a way to improve the robustness of our setup. In order to formalize the lead time trade-off (longer lead time leads to more independence, shorter lead time leads to more boosted simulations exceeding T_ref), we would need to be able to reliably estimate the spread as a function of lead time. While this is possible in theory (chaotic theory predicting an exponential growth initially, with a saturation to climatological values), there is no local guarantee that each event will follow this theoretical prediction, especially since the events considered here are extreme, and might therefore behave differently than a mean prediction. Indeed, we can see in Figure 6 that the spread per lead time is very different from one event to another. While it would be possible to train on the large amounts of data produced with boosting to predict a more accurate spread per lead time, this result would be dependent on the specifics of the event type (heat wave) and location (PNW).

In order to make this clear in the manuscript, we have added the following sentence in l. 478-482:

*"Note that one cannot reliably estimate the spread as a function of lead time, since each boosted simulation may not exactly follow the theoretically predicted ergodic growth speed, in particular due to the extreme nature of the parent events. This is evidenced by the large variation of growth exhibited between boosted simulations from different parents (see Figure 6). Therefore we cannot explicitly state the boundary lead times of this trade-off, in particular since any effort to do so would be both location- and variable-specific (heatwaves in the PNW region)."*

- 394-396: How do you explain this? Note that I wouldn't necessarily accept that the estimates from fitted GEV distributions are "naïve". This seems naïvely dismissive in my view. As I mentioned above, there might be reasons to think that a boosted sample of extremes from short simulations may estimate something different from the thing that can be estimated with annual block maxima.

We thank the reviewer for highlighting this. Indeed, the phrasing "GEV fit of the naive estimator from the reference 30-member Large Ensemble" is not correct and has been changed to "the GEV fit of the reference 30-member Large Ensemble".

What is methodologically done is to fit a GEV law to the data from the control period and test slices (shown as a black or blue line, with bootstrapped confidence intervals). Additionally, we calculate the return periods of the same data sets with the naive estimator (shown in black or blue dots), but these are separate operations and have separate purposes.

Since GEV return times are systematically biased for small sample sizes (which, one could argue, is the case for both the 30 and 10-member LE here, given the high return level in question, especially compared to the 4000-year PI control run and the lower return levels considered in the first experiment), this could be one argument to explain why the return periods are larger than for the boosting estimator, and why the

30-member LE estimates a larger return period than the 100-member LE. This is further discussed in discussion section 4.4.

- 404-405: Suddenly, there is discussion of an event "E1" without a definition of the event. (please improve the notation …).
  In addition to adding both E1 and ~E1 to the glossary, we have changed the sentence to: "One hypothesis that could explain this deviation is that the boosted simulations from the most extreme parent event, *hereafter denoted by* E1, are biased due to the intensity of the event."

- 420: How do we know that it is a more robust estimate, or that it more realistic than other estimates that suggest that the heat dome was a very low probability event?
  We say that the estimation is more robust, since the 30-member LE return times depend heavily on the inclusion of the most extreme event E1. The boosting estimator, however, provides similar results when all simulations from a specific parent are removed (see figure 9 for removing E1, and figure A7 for removing all the other parents).
  We have clarified this in the manuscript by changing the sentence to: "However, the boosting estimator can provide *return periods for extreme events like the 2021 PNW heatwave that depend less on E1, and that, importantly, stay finite*."

- 502-504: I would agree that extrapolation into the deep upper tail of the GEV distribution has many limitations, but the fundamental limitation is not uncertainty quantification via bootstrapping, as suggested here. Rather, it is the assumption of max-stability (the notion that block-maxima of samples of block-maxima again have GEV distributions with the same shape parameter). This implies that there will be no "surprises" in the unobserved parts of the tail that do not conform with the behaviour seen in the sampled parts of the distribution. I think this implicitly assumes that there is only one physical process that generates extremes at a given location – while in reality, it might be reasonable to think that the PNW heat dome event reflects the impact of a process that is distinct from the process (or collection of processes) that produces the annual maximum temperature event in the region in most years.
  We thank the reviewer for highlighting this. Here, we mention the overconfidence coming from bootstrapping, but also the systematic overestimation of return periods when fitting GEV distribution laws onto limited data, which ties more in with the reviewer's remarks.
  In order to highlight this further, we have changed the sentence to: "it might be that the *return periods are not biased by E1, but rather that the* GEV distribution fit is both overconfident (only capturing uncertainties from bootstrapping the limited sample, leading to a smaller confidence interval) and *systematically* overestimating return periods *due to limitations of the sample size at hand*."

  Additionally, while there are several processes that contribute to heatwaves, the typicality argument postulates that the more extreme events are, the more they are dynamically similar. This argument has been corroborated in several studies[7][8][9] (all three studies analyze heatwaves, and one demonstrates that this also holds for the 2021

PNW heatwave).

However, it is important to note that this typicality in the tail does not mean that there will be no "surprises", since there could exist an "extreme extreme" that is dynamically distinct from the simply extreme, but very similar to any other (extremely rare) event of the same extreme extreme magnitude. In this paper, for example, we find that the events E1, ~E1 seem to present a distinct atmospheric picture in comparison to that of the other events. This hints at a possible bimodality in the tail, which is discussed in the following paragraph (lines 624-632).

---

## Author Response (AR2)

Egusphere-2025-525: Estimating return periods for extreme events in climate models through Ensemble Boosting

Response to the reviewer's comments by Luna Bloin-Wibe, Robin Noyelle, Vincent Humphrey, Urs Beyerle, Reto Knutti and Erich Fischer.

**Co-editor decision: Publish subject to minor revisions**

The present paper presents a novel methodology that allows for the construction of ensembles that improve the estimation of the return period of extreme events by reducing the computing time needed to refine naive estimates of this return period. The paper presents a clear methodology and I hope that it will become a useful contribution for the determination of the return period of heat waves and other related extreme phenomena.

We have reviewed all minor details such as typos, missing co-authors and their affiliations, terminology, updates of data in tables, or updates of variables in equations. Additionally, we have supplemented our acknowledgment section by thanking the reviewers for their contributions.